# Mutual Multilinearity of Nonequilibrium Network Currents

**Sara Dal Cengio**$^{1,3\star}$, **Pedro E. Harunari**$^{2\dagger}$, **Vivien Lecomte**$^1$, and **Matteo Polettini**$^4$

**1** Université Grenoble Alpes, CNRS, LIPhy, FR-38000 Grenoble, France
**2** Department of Physics and Materials Science, University of Luxembourg, Campus Limpertsberg, 162a avenue de la Faïencerie, L-1511 Luxembourg (G. D. Luxembourg)
**3** Department of Physics, Massachusetts Institute of Technology, Cambridge, Massachusetts 02139, USA
**4** Via Gaspare Nadi 4, 40139 Bologna (Italy)

$\star$ saradc@mit.edu ,    $\dagger$ pedro.harunari@uni.lu

## Abstract

Continuous-time Markov chains have been successful in modelling systems across numerous fields, with currents being fundamental entities that describe the flows of energy, particles, individuals, chemical species, information, or other quantities. They apply to systems described by agents transitioning between vertices along the edges of a network (at some rate in each direction). It has recently been shown by the authors that, at stationarity, a hidden linearity exists between currents that flow along edges: if one controls the current of a specific 'input' edge (by tuning transition rates along it), any other current is a linear-affine function of the input current [**PRL 133, 047401 (2024)**]. In this paper, we extend this result to the situation where one controls the currents of several edges, and prove that other currents are in linear-affine relation with the input ones. Two proofs with distinct insights are provided: the first relies on Kirchhoff's current law and reduces the input set inductively through graph analysis, while the second utilizes the resolvent approach via a Laplace transform in time. We obtain explicit expressions for the current-to-current susceptibilities, which allow one to map current dependencies through the network. We also verify from our expression that Kirchhoff's current law is recovered as a limiting case of our mutual linearity. Last, we uncover that susceptibilities can be obtained from fluctuations when the reference system is originally at equilibrium.

# 1 Introduction

In nonequilibrium Markov chains, the net number of times an edge is traversed is called a current, constituting a key observable. Close to equilibrium, currents (of any kind) flow proportionally to conjugated forces and therefore respond linearly to one another. This is the tenet of linear thermodynamics, which grounds on the notion of conjugated observables [1] and the varied fluctuation-dissipation relations that come with it [2–4]. Far from equilibrium, currents are generally nonlinear functions of the conjugated forces and therefore nonlinear functions of one another, and fluctuation-dissipation relations break down.

Historically, only a few relations are known to hold arbitrarily far from equilibrium, such as the fluctuation theorems [5–9], the thermodynamic uncertainty relation [10, 11], and extensions of the fluctuation-dissipation relation [12–16]. Recently, new results on the response of observables have started to surface [17–29], which include fluctuation-response relations [30, 31], the connection between responses and correlations of dynamical events [32], and the mutual linearity of currents [33]. In this latter study [33], we have shown that any two stationary currents always satisfy a linear-affine relation between each other, even arbitrarily far from equilibrium, when the transition rates of a single edge are controlled. It has been conjectured that a similar property might hold when more edges are controlled [34]. In the present work, we prove that a linear-affine relation among stationary currents can indeed be established even when the transition rates along multiple edges are simultaneously controlled. We explore the details and conditions for its validity, as well as the recovery of Kirchhoff's current law and connections to linear response theory.

A special case of linear relation among stationary currents holds in the form of the Kirchhoff current law (KCL), which is a central result in Schnakenberg's geometrical theory of Markov chains [35–37]. Therein, a Markov chain is represented as a network (see Fig. 1), where each vertex corresponds to a state of the Markov chain and an edge is drawn between two vertices whenever a transition between the corresponding states is possible. Using geometrical [36] or algebraic tools [38], it is possible to identify a subset of edges whose removal from the network results in a spanning tree, i.e. a tree that connects all vertices (see Fig. 1). The subset of edges that defines a spanning tree is not unique; it is referred to as a *fundamental set*, and once it is fixed, each element of this set is called a *chord*. The number of chords in a fundamental set is fixed by the network's topology and is known as the cyclomatic number $n_c$. Notably, for a given fundamental set, each chord can be uniquely associated with a specific cycle in the network.

According to Kirchhoff current law, any stationary edge current $J_e$ can be expressed as a superposition of the stationary currents $J_i$ flowing along the chords of a fundamental set, as:

$$J_e = \sum_{i=1}^{n_c} \gamma_{e,i} J_i \tag{1}$$

where $\gamma_{e,i} = \pm 1, 0$ are algebraic coefficients specifying whether edge $e$ belongs to cycle $i$ and the relative orientation within it. Eq. (1) is the result of the conservation of probability, which imposes constraints on the value that currents can take at steady-state. It tells that if an observer were to control or fix the value of all $n_c$ chord currents, all other currents would be determined by these values through the linear topological constraints of Eq. (1). From a thermodynamic viewpoint, the chord currents $J_i$'s can be viewed as the independent degrees of freedom at steady state associated with thermodynamic forces (or affinities) $a_i$ which contribute independently to the stationary entropy production. As such, Eq. (1) underpins a complete thermodynamic description of stationary currents, which have proved valuable both from a practical and a conceptual viewpoint [38–44].

Recently, a different form of linearity among stationary currents has been established by us

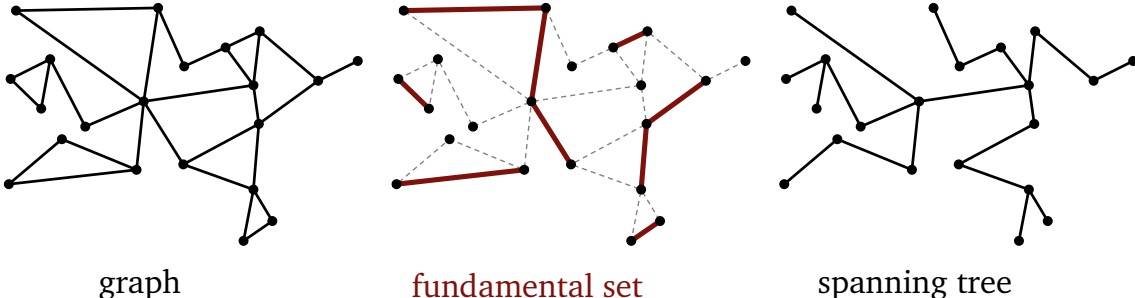

graph                    fundamental set                  spanning tree

Figure 1: *Illustration of central graph-theoretical notions.* Starting from a connected graph (left panel), a set of edges is identified in red (middle panel) whose removal results in a spanning tree (right panel), a connected graph without cycles that contains all vertices. Adding back one chord to the spanning tree creates exactly one cycle.

in Ref. [33]. Therein, we ask the question of how stationary currents respond to manipulations of the transition rates of a single edge current, which we term *input current*. We proved that any current of the Markov process $J_e$ can be expressed as a linear-affine function of a single input current $J_i$, as:

$$J_e = J_e^{\searrow i} + \lambda_{e \leftarrow i}^i J_i. \tag{2}$$

In contrast to KCL, here the scalar coefficients $J_e^{\searrow i}$ and $\lambda_{e \leftarrow i}^i$ are functions on the transition rates of the Markov chains, excluding the transition rates of the input current itself. As such, they depend both on topological and dynamical properties of the Markov process. In particular, $J_e^{\searrow i}$ is the value of the current $J_e$ in the Markov chain where the input edge is removed. Interestingly, the dependencies of $J_e^{\searrow i}$ and $\lambda_{e \leftarrow i}^i$ on the transition rates can be made explicit using the technique of spanning tree polynomials [33,45]. For simplicity, here and in the rest of the paper, we use the term linearity for linear-affine relations like Eq. (2). The affine coefficient is assumed to be present unless otherwise stated.

These two forms of mutual linearity among currents, both valid far from equilibrium, are utterly different in nature. On one hand, KCL states that any stationary current can be expressed as a linear combination of a full set of chord currents, with simple coefficients $\in \{-1, 0, +1\}$, and is the consequence of probability conservation. On the other hand, Eq. (2) reveals a linear-affine relation between any stationary current and one input current, when only the rates of the latter are controlled.

In the present paper, we address the question of what is the fate of the linearity in Eq. (2) when several transitions rates, belonging to different edges, are simultaneously controlled. We demonstrate that linearity persists even when multiple $n \leq n_c$ chord currents are controlled, and obtain explicit expressions for the current-current susceptibilities. We check that, as expected, such expressions yield back KCL, Eq. (1), in the case $n = n_c$. In this regard, Eqs. (1)-(2) can be seen as two limiting cases of a more general form of mutual multilinearity (MML) governing the system's behavior to changes in transition rates (see Fig. 2). We will provide two different proofs, one based on a direct graph-theoretic analysis and one based on the Laplace transform analysis. The strategy for the graph-theoretic proof is a "top-down" induction with a *seed* treating the case of one input edge less than the cyclomatic number, and a *step* treating the case of one input edge less than in the previous step. The above result holds for arbitrary graph geometry, arbitrary choices of transition rates and arbitrary variations of the input rates.

The plan of the paper is as follows. In Sec. 2 we introduce the framework of Markov processes and state our main result. Sections 3 and 4 are devoted to two different proofs, which provide different insights. The graph-theoretical proof explicitly employs the Kirchhoff

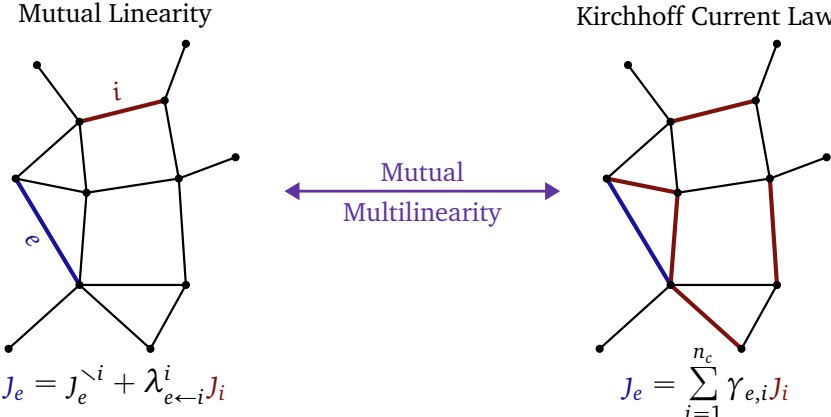

Figure 2: *Summarizing two types of mutual linearity among nonequilibrium station-ary currents.* (Left) Any stationary current $J_e$ (blue) in the graph can be represented as a linear-affine function of an "input" current $J_i$ (red) when the input edge $i$ is controlled; the scalar coefficients $J_e^{\searrow i}$ and $\lambda_{e\leftarrow i}^i$ depend both on dynamical and topo-logical properties of the Markov process. This result expresses the mutual linearity of currents, derived in Ref. [33]. (Right) Kirchhoff's current law: A fundamental set of edges (red) whose removal leaves a connected graph without cycles is fixed. The stationary current of any other edge (blue) can always be represented as a lin-ear combination of the currents along the red edges (including when these edges are controlled). These choices of edges are not unique, and the coefficients $\gamma_{e,i}$ are either 0 or $\pm 1$ depend solely on the graph topology. In this paper, we address the intermediate situation in which more than one edge is controlled, but KCL does not hold yet.

Current Law, Eq. (1), while the Laplace transform proof allows for a generalization at finite time, and provides connection with the notion of current-to-current susceptibility [33]. Finally, in Sec. 5, we show how to recover KCL as a limiting case from MML, and make connection with the framework of linear response by introducing the notion of affinities. Remarkably, we show that the transport coefficients commonly defined close to equilibrium in fact describe the current-current relation arbitrarily far from equilibrium, provided that the reference transition rates fulfill detailed balance. Finally, in Sec. 6 we draw our conclusions and outline future perspectives.

Remark: Note that during the writing of this work, F. Khodabandehlou and coworkers have obtained analogous results for multigraphs [46] using different methods; our contribution sheds light on the conditions ensuring that the susceptibilities are well-defined and unique (and on the time-dependent case, in Laplace space).

## 2 Setup and statement of the main result

Consider a connected graph $\mathscr{G}$ with vertices $\mathrm{x} \in \mathscr{X}$ and edges $e \in \mathscr{E}$. We assume no multiple edges between two vertices. To each edge we assign a reference forward orientation $+e$ (or simply $e$) from a source to a target vertex. The incidence matrix $\mathbf{S}$ has entries $\mathbf{S}_{\mathrm{x}e}$ equal to $-1$ if $\mathrm{x}$ is the source of $e$, $+1$ if $\mathrm{x}$ is the target of $e$ and 0 otherwise, for $\mathrm{x} \in \mathscr{X}$ and $e \in \mathscr{E}$. We denote $-e$ the edge with opposite orientation and $\overleftarrow{\mathscr{E}}$ the set of edges oriented backwards. We assign positive weights $r : \mathscr{E} \cup \overleftarrow{\mathscr{E}} \to \mathbb{R}_+$ to forward and backward directions of an edge. For notational simplicity we now define $\overline{\mathscr{E}} = \mathscr{E} \cup \overleftarrow{\mathscr{E}}$. Let $r_{\mathscr{A}}$ be the rates of a subset $\mathscr{A} \cup \overleftarrow{\mathscr{A}} \subseteq \overline{\mathscr{E}}$.

Consider the stationary continuous-time Markov chain having $r$ as transition rates. Let $\mathbf{R}$ be the rate matrix with entries $R_{\mathrm{xy}} = r_{\mathrm{x}\leftarrow\mathrm{y}} - \delta_{\mathrm{x,y}} \sum_{\mathrm{z}\in\mathcal{X}} r_{\mathrm{z}\leftarrow\mathrm{x}}$. The probability $\mathbf{p}(t) = (p_{\mathrm{x}}(t))_{\mathrm{x}\in\mathcal{X}}$ of being at given vertices at time $t$, given an initial probability $\mathbf{p}(0)$, evolves according to the master equation

$$\dot{\mathbf{p}}(t) = \mathbf{R}\mathbf{p}(t). \tag{3}$$

Let $\boldsymbol{\pi} = (\pi_{\mathrm{x}})_{\mathrm{x}\in\mathcal{X}}$ be the normalized stationary distribution satisfying $\mathbf{R}\boldsymbol{\pi} = 0$, which by the above assumptions is unique. For simplicity, we also assume $r_e \neq 0 \Leftrightarrow r_{-e} \neq 0$ (weak reversibility).

The main object in our study are the stationary currents, which for the oriented edge $e = \mathrm{x} \leftarrow \mathrm{y}$ are given by

$$J_e = J_{\mathrm{x}\leftarrow\mathrm{y}} = r_{\mathrm{x}\leftarrow\mathrm{y}}\pi_{\mathrm{y}} - r_{\mathrm{y}\leftarrow\mathrm{x}}\pi_{\mathrm{x}}. \tag{4}$$

A straightforward consequence of the master equation is Kirchhoff's Current Law (KCL),

$$\sum_{e\in\mathcal{E}} \mathbf{S}_{\mathrm{x},e} J_e = 0 \quad \forall \mathrm{x} \tag{5}$$

stating that the in-and-out stationary currents balance at any vertex $\mathrm{x}$. An alternative way to express KCL makes use of the notion of spanning tree. In graph theory, a spanning tree is a minimal set of edges that connects all vertices (and thus contains no cycles). A consequence of Eq. (5) is that for any spanning tree $\mathcal{T}$ there exist unique coefficients $\gamma_{e,i} \in \{0, -1, +1\}$, not dependent on the rates, such that

$$J_e = \sum_{i\in\mathcal{E}\setminus\mathcal{T}} \gamma_{e,i} J_i \tag{6}$$

with the sum running over the set of edges not belonging to $\mathcal{T}$. These constitute a set of chord edges, and the input currents along them are said to be a fundamental set. Its cardinality $n_c$, the cyclomatic number, is the dimension of the kernel of the stoichiometric matrix $\mathbf{S}$. The vectors $\boldsymbol{\gamma} = (\gamma_{e,i})_{e\in\mathcal{E}}$ are null vectors of the incidence matrix, and they can be represented as simple oriented cycles [36].

In this work, we generalize Eq. (6) to the case where the $n < n_c$ input currents are a subset of some fundamental set, a condition that we term *admissibility* (see Fig. 3).

Now consider $n$ input currents along admissible edges $\mathcal{I} = (i_1, ..., i_n)$. Our main result is the following. For every edge $e \in \mathcal{E}$ there exist, and are uniquely defined, a coefficient $J_e^{\searrow\mathcal{I}}$ and $n$ coefficients $\lambda_{e\leftarrow i}^{\mathcal{I}}$ (for $i \in \mathcal{I}$), independent of the $2n$ rates in $\boldsymbol{r}_{\mathcal{I}}$, such that when varying such rates

$$J_e(\boldsymbol{r}_{\mathcal{I}}) = J_e^{\searrow\mathcal{I}} + \sum_{i\in\mathcal{I}} \lambda_{e\leftarrow i}^{\mathcal{I}} J_i(\boldsymbol{r}_{\mathcal{I}}). \tag{7}$$

Notice that in general the above coefficients depend on the rest of the rates $\boldsymbol{r}_{\mathcal{E}\setminus\mathcal{I}}$.

Before proceeding, we discuss some relevant properties of this relation and its constituents. Both coefficients are unique. The affine coefficient $J_e^{\searrow\mathcal{I}}$ represents the current along edge $e$ when every edge in $\mathcal{I}$ is removed from the graph. The linear coefficient $\lambda_{e\leftarrow i}^{\mathcal{I}}$ of $J_i(\boldsymbol{r}_{\mathcal{I}})$ represents a current-current susceptibility when considering the $n$ input currents (associated to the edges in $\mathcal{I}$). We show that $\lambda_{e\leftarrow i}^{\mathcal{I}}$ is equal to the one-input-current susceptibility in the graph obtained after removing edges $\mathcal{I}\setminus\{i\}$ (whose expression in terms of spanning tree polynomials was obtained in [33]). Furthermore, when modifying rates starting from a detailed-balance reference state, linear coefficients can be sampled from equilibrium fluctuations, as discussed in Sec. 5.2. The mutual multilinearity (MML) of Eq. (7) reduces to Eq. (2) in the case of one

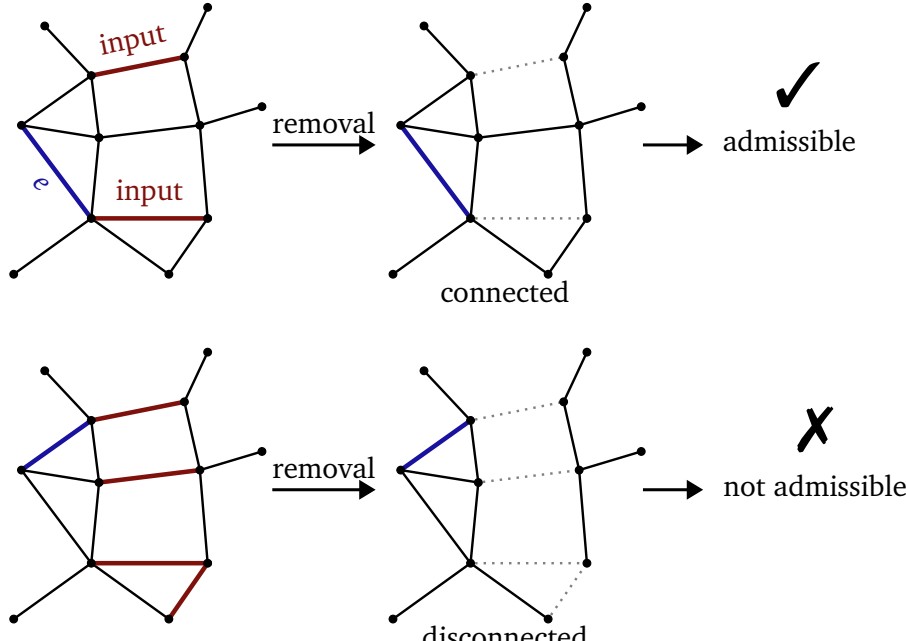

Figure 3: *Illustration of the admissibility concept.* Top: The removal of the input edges results in a connected graph, satisfying the criterion for admissibility. Bottom: The resulting graph is disconnected thus the input set is not admissible.

input current and to KCL when $n = n_c$. In this latter case, the current-current susceptibility $\lambda^{\mathcal{I}}_{e \leftarrow i}$, which can be obtained explicitly using graph-theoretical analysis, reduces to an algebraic coefficient consistent with KCL, as shown in Sec. 5.1. We stress that Eq. (7) is a property of stationary states, likewise is KCL; a generalization of Eq. (7) to finite-time in the Laplace domain is discussed in Sec. 4.

## 3   Graph-theoretic steady-state proof

We proceed by induction by proving the result for $n = n_c - 1$ and then by decreasing recursively the cardinality $n$ of the admissible set of edges by steps of 1.

Notice that an admissible set of input currents is any set such that removal of the corresponding edges does not disconnect the graph (otherwise their complement could not contain a spanning tree, which by definition connects all vertices); vice-versa, the complement of any set of edges whose removal does not disconnect the graph contains a spanning tree, whose complement is a chord set. See Figure 3.

### 3.1   Proof of inductive seed

Consider an admissible set $\mathcal{I}$ of $n_c - 1$ edges. By admissibility, one can find an edge $\alpha \in \mathscr{E} \smallsetminus \mathcal{I}$ such that $\mathcal{I}_\alpha := \mathcal{I} \cup \{\alpha\}$ is a fundamental set of chords, i.e. such that $\mathscr{T} = \mathscr{E} \smallsetminus \mathcal{I}_\alpha$ is a spanning tree. From KCL, we know that any edge currents $J_e$ with $e \in \mathscr{T}$ can be expressed as a linear combination (with coefficients in $\{0, \pm1\}$) of the currents $(J_i)_{i \in \mathcal{I}_\alpha}$. Our goal is to show that currents $(J_e)_{e \in \mathscr{E} \smallsetminus \mathcal{I}}$ are linear-affine functions of the input currents $(J_i)_{i \in \mathcal{I}}$, with coefficients that depend only on the rates $(r_e)_{e \in \mathscr{E} \smallsetminus \mathcal{I}}$. From KCL, to show this, it is sufficient to prove that $J_\alpha$ can itself be represented as a linear-affine combination of the currents $(J_i)_{i \in \mathcal{I}}$ with coefficients that depend only on the rates $(r_e)_{e \in \mathscr{E} \smallsetminus \mathcal{I}}$.

Our strategy is to produce two linear relations between the stationary probability on one particular vertex $x_0$ and the stationary currents, taking care that the coefficients depend appropriately on the rates. We can then use such relations to eliminate the stationary probability and one of the currents, and deploy Kirchhoff's Current Law.

A useful observation to simplify the analysis below is that, for a fixed spanning tree $\mathscr{T}$ and a chord $\alpha$ (that we pick as defined above), there always exists a parametrization of the rates in terms of:

- a so-called potential $\boldsymbol{u} = (u_x)_{x \in \mathscr{X}}$ (defined up to an arbitrary ground potential), that depends only on the rates $(r_e)_{e \in \mathscr{T}}$,

- a so-called cycle affinity $a_\alpha$ (associated to the oriented cycle $\mathscr{C}_\alpha$ generated when adding $\alpha$ to $\mathscr{T}$), that depends only on the rates $(r_e)_{e \in \mathscr{T}}$ and on the two rates $r_\alpha$

such that (see App. A):

$$\frac{r_{x \leftarrow y}}{r_{y \leftarrow x}} = e^{u_y - u_x}, \qquad \text{for } x \leftarrow y \in \overline{\mathscr{T}}, \tag{8}$$

$$\frac{r_{x \leftarrow y}}{r_{y \leftarrow x}} = e^{u_y - u_x \pm a_\alpha}, \qquad \text{for } x \leftarrow y = \pm \alpha. \tag{9}$$

With this in mind, we turn to the current-probability relation that we will use below and that follows straightforwardly from the definition of stationary current Eq. (4). For $x \leftarrow y \in \overline{\mathscr{E}}$:

$$\pi_x = \frac{r_{x \leftarrow y} \pi_y + J_{y \leftarrow x}}{r_{y \leftarrow x}}. \tag{10}$$

This suggests to define the following pseudo-potential $\boldsymbol{\mu}$ and pseudo-affinity $\boldsymbol{h}$

$$\mu_x = e^{u_x} \pi_x \quad \text{for } x \in \mathscr{X}, \qquad \text{and} \qquad h_{y \leftarrow x} = e^{u_x} \frac{J_{y \leftarrow x}}{r_{y \leftarrow x}} = e^{u_y} \frac{J_{y \leftarrow x}}{r_{x \leftarrow y}} \quad \text{for } x \leftarrow y \in \overline{\mathscr{T}} \tag{11}$$

such that, from Eqs. (8) and (10),

$$\mu_x - \mu_y = h_{y \leftarrow x} \qquad \forall x \leftarrow y \in \overline{\mathscr{T}}. \tag{12}$$

Note that $h_{x \leftarrow y} = -h_{y \leftarrow x}$ as easily checked, provided $x \leftarrow y$ belongs to $\overline{\mathscr{T}}$. The addition of the chord $\alpha$ to the spanning tree $\mathscr{T}$ produces an oriented cycle $\mathscr{C}_\alpha$ with $k+1$ vertices $(x_0, x_1, \ldots, x_k)$ and $k+1$ edges $(x_0 \leftarrow x_1, x_1 \leftarrow x_2, \ldots, x_{k-1} \leftarrow x_k, \alpha = x_k \leftarrow x_0)$. On the chord $\alpha$, the application of Eqs. (9) and (10) yields

$$\mu_{x_0} - e^{-a_\alpha} \mu_{x_k} = h_\alpha \qquad \text{with} \quad h_\alpha = e^{u_{x_0}} \frac{J_\alpha}{r_\alpha}. \tag{13}$$

The field $h$ is a pseudo-affinity in the sense that, for a chord like $\alpha$, we have an extra prefactor $e^{-a_\alpha}$ in Eq. (13) compared to Eq. (12), and we also have $h_{-\alpha} = -e^{a_\alpha} h_\alpha$ as easily checked.

The first linear relation is found as follows. Starting from edge $x_0 \leftarrow x_1$, we propagate Eq. (12) around the cycle $\mathscr{C}_\alpha$ using a telescopic sum:

$$\mu_{x_k} - \mu_{x_0} = \sum_{0 \leq p < k} \left( \mu_{x_{p+1}} - \mu_{x_p} \right) = \sum_{0 \leq p < k} h_{x_p \leftarrow x_{p+1}}. \tag{14}$$

On the last edge $\alpha = x_k \leftarrow x_0$ of the cycle we use, from Eq. (13), $\mu_{x_k} = e^{a_\alpha} \mu_{x_0} - e^{a_\alpha} h_\alpha$ to get

$$\left( e^{a_\alpha} - 1 \right) \mu_{x_0} = e^{a_\alpha} h_\alpha + \sum_{0 \leq p < k} h_{x_p \leftarrow x_{p+1}}. \tag{15}$$

Notice that if the affinity vanishes ($a_\alpha = 0$), the above expression already yields a linear relation among the currents. We investigate this further below, but for now we assume that $a_\alpha \neq 0$.

The second linear relation comes from the normalization $1 = \sum_{y \in \mathscr{X}} \pi_y = \sum_{y \in \mathscr{X}} e^{-u_y} \mu_y$. We root the spanning tree $\mathscr{T}$ in $x_0$ by orienting all its edges towards $x_0$. For any vertex $y \neq x_0$, Eq. (12) implies that $\mu_y$ can be found by integrating the pseudo-affinity $h$ along the unique path along $\mathscr{T}$ between $y$ and $x_0$. Defining the vector $\boldsymbol{h} = (h_e)_{e \in \mathscr{T}}$ (with $|X| - 1$ entries), this integration procedure is encoded by a $(|X|-1) \times (|X|-1)$ path matrix $\mathbf{P}$, with entries $\in \{0, 1\}$, such that:

$$\mu_y = (\mathbf{P}\boldsymbol{h})_y + \mu_{x_0} \quad \text{for all } y \neq x_0 \, . \tag{16}$$

The matrix $\mathbf{P}$ can be found by inverting a "core" submatrix of the incidence matrix of the graph (see for instance Ref. [38]). Inserting Eq. (16) in the normalization condition, we arrive at

$$\mu_{x_0} = \frac{1 - \sum_{\substack{y \in \mathscr{X} \\ y \neq x_0}} e^{-u_y} (\mathbf{P}\boldsymbol{h})_y}{\sum_{y \in \mathscr{X}} e^{-u_y}} \, . \tag{17}$$

From KCL, the currents $(J_e)_{e \in \mathscr{T}}$ are a linear combination of the chord currents $(J_i)_{i \in \mathcal{I}_\alpha}$, with coefficients in $\{0, \pm 1\}$. From Eq. (11), this implies that there exists a $(|\mathscr{X}|-1) \times (n_c - 1)$ matrix $\mathbf{K}$ and a vector $\boldsymbol{k}^\alpha$ with $|\mathscr{X}| - 1$ entries, both depending on the rates $\boldsymbol{r}_\mathscr{T}$, such that

$$\boldsymbol{h} = \mathbf{K}\boldsymbol{J}_\mathcal{I} + J_\alpha \boldsymbol{k}^\alpha \, , \tag{18}$$

where $\boldsymbol{J}_\mathcal{I} = (J_i)_{i \in \mathcal{I}}$ is the vector of input currents. From this and Eq. (13), we insert Eq. (17) into Eq. (15) so as to eliminate $\mu_{x_0}$. We arrive at a linear-affine relation between $J_\alpha$ and the input currents $\boldsymbol{J}_\mathcal{I}$:

$$\frac{e^{a_\alpha} - 1}{\sum_{y \in \mathscr{X}} e^{-u_y}} \left[ 1 - \sum_{\substack{y \in \mathscr{X} \\ y \neq x_0}} e^{-u_y} (\mathbf{P}\mathbf{K}\boldsymbol{J}_\mathcal{I})_y \right] - \boldsymbol{c} \cdot \mathbf{K}\boldsymbol{J}_\mathcal{I}$$

$$= \left[ \frac{e^{a_\alpha + u_{x_0}}}{r_\alpha} + \boldsymbol{c} \cdot \boldsymbol{k}^\alpha + \frac{e^{a_\alpha} - 1}{\sum_{y \in \mathscr{X}} e^{-u_y}} \sum_{\substack{y \in \mathscr{X} \\ y \neq x_0}} e^{-u_y} (\mathbf{P}\boldsymbol{k}^\alpha)_y \right] J_\alpha \, , \tag{19}$$

where $\boldsymbol{c}$ is the vector of entries in $\{0, 1\}$ such that $\sum_{0 \leq p < k} h_{x_p \leftarrow x_{p+1}} = \boldsymbol{c} \cdot \boldsymbol{h}$. The prefactor of $J_\alpha$ in the r.h.s. of Eq. (19), which is a function of the rates $\boldsymbol{r}_{\mathscr{E} \setminus \mathcal{I}}$, cannot cancel. Indeed, if it were so, the l.h.s. of Eq. (19) would be zero $\forall \boldsymbol{r}_\mathcal{I}$, implying a linear-affine interdependence of the input currents $\boldsymbol{J}_\mathcal{I}$, with coefficients depending on $\boldsymbol{r}_{\mathscr{E} \setminus \mathcal{I}} = \boldsymbol{r}_{\mathscr{T} \cup \{\alpha\}}$ — but, from the independence lemma proved in App. B, this is impossible, since then the affine contribution $(e^{a_\alpha} - 1)/\sum_{y \in \mathscr{X}} e^{-u_y}$ would be 0; which is absurd.

To conclude, we infer from Eq. (19) that the chord current $J_\alpha$ is a linear-affine combination of the input currents $\boldsymbol{J}_\mathcal{I}$ with coefficients that are well-defined functions of the rates $\boldsymbol{r}_{\mathscr{E} \setminus \mathcal{I}}$. Using KCL, this shows that any current $(J_e)_{e \in \mathscr{E} \setminus \mathcal{I}}$ is a linear-affine combination of the input currents $\boldsymbol{J}_\mathcal{I}$ with coefficients that are functions of the rates $\boldsymbol{r}_{\mathscr{E} \setminus \mathcal{I}}$, which ends the proof of the initialization step of the recursion.

*Remark*: the "cycle equilibrium" case when $a_\alpha = 0$ is even simpler, and does not involve the normalization condition. Eq. (15) indeed then reads $h_\alpha = -\boldsymbol{c} \cdot \boldsymbol{h}$ which, using Eq. (18), yields

$$\left( \frac{e^{u_{x_0}}}{r_\alpha} + \boldsymbol{c} \cdot \boldsymbol{k}^\alpha \right) J_\alpha = -\boldsymbol{c} \cdot \mathbf{K}\boldsymbol{J}_\mathcal{I} \, , \tag{20}$$

which, following the same line of arguments as above, allows one to conclude.

## 3.2 Proof of inductive step

By assuming that mutual linearity holds for a set of $n < n_{\mathrm{c}}$ input edges, we want to prove that it also holds for $n - 1$.

Consider $\mathcal{I}$ a set of $n-1$ input edges under the single assumption that its removal does not disconnect the network, i.e. it is admissible. Our aim is to show that the current $J_e$ of any edge $e \in \mathscr{E} \setminus \mathcal{I}$ can be decomposed in a linear-affine way on the $J_{i \in \mathcal{I}}$ with coefficients independent on the rates $\boldsymbol{r}_{\mathcal{I}}$. To show this, we distinguish two cases.

Let us first assume that $e$ is such that $\mathcal{I} \cup \{e\}$ is also admissible, and define $e_2 := e$. Since $|\mathcal{I}| \le n_{\mathrm{c}} - 2$, it is always possible to find another edge $e_1 \notin \mathcal{I} \cup \{e_2\}$ such that both $\mathcal{I}_1 := \mathcal{I} \cup \{e_1\}$ and $\mathcal{I}_2 := \mathcal{I} \cup \{e_2\}$ are admissible. Since we assumed mutual linearity holds at the level of $n$ input edges, the (output) current supported by an edge $e_2 \notin \mathcal{I}_1$ can be expressed as

$$J_{e_2} = J_{e_2}^{\grave{}\,\mathcal{I}_1}(\boldsymbol{r}_{\mathscr{E} \setminus \mathcal{I}_1}) + \sum_{i_1 \in \mathcal{I}_1} \lambda_{e_2 \leftarrow i_1}^{\mathcal{I}_1}(\boldsymbol{r}_{\mathscr{E} \setminus \mathcal{I}_1}) J_{i_1} \,, \tag{21}$$

where the right-hand side is linear-affine on the input currents with coefficients that depend on the rates of edges in $\mathscr{E} \setminus \mathcal{I}_1$. Similarly, for the edge $e_1$, because $\mathcal{I}_2$ is admissible, we can write

$$J_{e_1} = J_{e_1}^{\grave{}\,\mathcal{I}_2}(\boldsymbol{r}_{\mathscr{E} \setminus \mathcal{I}_2}) + \sum_{i_2 \in \mathcal{I}_2} \lambda_{e_1 \leftarrow i_2}^{\mathcal{I}_2}(\boldsymbol{r}_{\mathscr{E} \setminus \mathcal{I}_2}) J_{i_2} \,. \tag{22}$$

We can now replace Eq. (22) on the right-hand side of (21) and collect $J_{e_2}$, yielding

$$\begin{aligned}
\Big[ 1 - \lambda_{e_2 \leftarrow e_1}^{\mathcal{I}_1}(\boldsymbol{r}_{\mathscr{E} \setminus \mathcal{I}_1}) \lambda_{e_1 \leftarrow e_2}^{\mathcal{I}_2}(\boldsymbol{r}_{\mathscr{E} \setminus \mathcal{I}_2}) \Big] J_{e_2} = \\
J_{e_2}^{\grave{}\,\mathcal{I}_1}(\boldsymbol{r}_{\mathscr{E} \setminus \mathcal{I}_1}) + \lambda_{e_2 \leftarrow e_1}^{\mathcal{I}_1}(\boldsymbol{r}_{\mathscr{E} \setminus \mathcal{I}_1}) J_{e_1}^{\grave{}\,\mathcal{I}_2}(\boldsymbol{r}_{\mathscr{E} \setminus \mathcal{I}_2}) \\
+ \sum_{i \in \mathcal{I}} \Big[ \lambda_{e_2 \leftarrow i}^{\mathcal{I}_1}(\boldsymbol{r}_{\mathscr{E} \setminus \mathcal{I}_1}) + \lambda_{e_2 \leftarrow e_1}^{\mathcal{I}_1}(\boldsymbol{r}_{\mathscr{E} \setminus \mathcal{I}_1}) \lambda_{e_1 \leftarrow i}^{\mathcal{I}_2}(\boldsymbol{r}_{\mathscr{E} \setminus \mathcal{I}_2}) \Big] J_i \,.
\end{aligned} \tag{23}$$

The multiplicative factor $1 - \lambda_{e_2 \leftarrow e_1}^{\mathcal{I}_1}(\boldsymbol{r}_{\mathscr{E} \setminus \mathcal{I}_1}) \lambda_{e_1 \leftarrow e_2}^{\mathcal{I}_2}(\boldsymbol{r}_{\mathscr{E} \setminus \mathcal{I}_2})$ cannot vanish (for any value of the rates), as shown in App. C, which proves the multilinearity of $J_{e_2}$ for the input set $\mathcal{I}$. Indeed from Eq. (23), $J_{e_2}$ is expressed as a linear-affine combination of the currents $J_{i \in \mathcal{I}}$ with coefficients depending only on the rates of the transitions in $(\mathscr{E} \setminus \mathcal{I}_1) \cup (\mathscr{E} \setminus \mathcal{I}_2) = \mathscr{E} \setminus \mathcal{I}$.

The argument works for any $e_2$ such that $\mathcal{I} \cup \{e_2\}$ is admissible. Consider now instead an edge $e$ such that removing $\mathcal{I} \cup \{e\}$ disconnects the network. We can fix a full set of chords that includes $\mathcal{I}$ (which is always possible since $\mathcal{I}$ is admissible). Since the removal of $e$ and $\mathcal{I}$ disconnect the network, by definition, $e$ belongs to a cocycle of the network [38]. If it is the only component of the cocycle, it means that $e$ is a bridge and the current $J_e$ is always zero. If instead the cocycle contains other edges, these are chords forming a subset $\mathcal{I}' \subseteq \mathcal{I}$ (because removing $\mathcal{I} \cup \{e\}$ disconnects the network). Then as a property of cocycles, the current $J_e$ can be written as a linear combination of the $J_{i' \in \mathcal{I}'}$, with coefficients $\pm 1$.

To summarize, in the second case where $\mathcal{I} \cup \{e\}$ is not admissible, the standard KCL directly gives the result (with rate-independent linear-affine coefficients), while in the first case ($\mathcal{I} \cup \{e\}$ admissible) the coefficients do depend on the rates and the derivation did require the recursion hypothesis at step $n$, completing the inductive step.

## 4 Laplace-space finite-time proof

MML in Eq. (7) is a property of steady state: already with one input current, the study of simple examples shows that it does not hold if we were to replace stationary currents by finite-time

ones [33]. Similarly, with $n_c$ input currents MML reduces to KCL (see Sec. 5.1), which only holds for stationary currents.

In this section, we show how one can generalize Eq. (7) by going to the Laplace domain, introducing a frequency variable $\varsigma$ conjugated to time $t$. By sending $\varsigma$ to zero (equivalent to sending $t$ to infinity), this provides another proof of Eq. (7) with an explicit expression of the current-current susceptibility $\lambda^{\mathcal{I}}_{e\leftarrow i}$. Interestingly, the proof offers a physical representation of such a susceptibility: $\lambda^{\mathcal{I}}_{e\leftarrow i}$ is equal to the susceptibility $\tilde{\lambda}^{i}_{e\leftarrow i}$ for one input current (along $i$) and one output current (along $e$) in a modified graph where every edge of $\mathcal{I}$ (apart from $i$) is removed. Such a representation is also valid at non-zero $\varsigma$. This allows one to represent $\lambda^{\mathcal{I}}_{e\leftarrow i}$ using tree ensembles, using results that we derived in Ref. [33].

## 4.1 Settings

The case of non-stationary currents can be understood by turning to the frequency domain, as follows. The probability distribution at time $t$ is the solution $\mathbf{p}(t) = \exp(t\mathbf{R})\mathbf{p}(0)$ to the master equation, given an initial probability distribution $\mathbf{p}(0)$. Defining the Laplace transform $\hat{\mathbf{p}}(\varsigma) = \int_0^\infty dt\, e^{-\varsigma t} \mathbf{p}(t)$ (and similarly for other functions of time), one arrives at the expression $\hat{\mathbf{p}}(\varsigma) = (\varsigma\mathbf{1} - \mathbf{R})^{-1}\mathbf{p}(0)$. Both the Laplace transform $\hat{\mathbf{p}}(\varsigma)$ and the resolvent $(\varsigma\mathbf{1} - \mathbf{R})^{-1}$ are defined for $\varsigma \in \mathbb{C} \setminus \mathrm{Sp}\,\mathbf{R}$, i.e. for complex numbers not belonging to the spectrum of $\mathbf{R}$.

We first prove a Laplace-domain generalization of MML:

$$\hat{\jmath}_e(\boldsymbol{r}_{\mathcal{I}}, \varsigma) = \hat{\jmath}_e^{\backslash \mathcal{I}}(\varsigma) + \sum_{i \in \mathcal{I}} \hat{\lambda}^{\mathcal{I}}_{e\leftarrow i}(\varsigma)\, \hat{\jmath}_i(\boldsymbol{r}_{\mathcal{I}}, \varsigma) \tag{24}$$

and then analyze the $\varsigma \to 0$ limit in Sec. 4.3 where we recover Eq. (7) (holding at stationarity $t \to \infty$).

## 4.2 Proof of mutual multilinearity

Proving Eq. (24) is equivalent to show the existence of constants $\hat{\lambda}^{\mathcal{I}}_{e\leftarrow i}(\varsigma)$'s such that the gradient of both sides w.r.t. the $2n$ variables $\boldsymbol{r}_{\mathcal{I}}$ is the same. In other words, denoting $\hat{\boldsymbol{\lambda}}^{\mathcal{I}}(\varsigma)$ the vector of $n$ components $\big(\hat{\boldsymbol{\lambda}}^{\mathcal{I}}(\varsigma)\big)_i = \hat{\lambda}^{\mathcal{I}}_{e\leftarrow i}(\varsigma)$ for $i \in \mathcal{I}$, we have to show that

$$\exists \hat{\boldsymbol{\lambda}}^I(\varsigma) : \forall k \in \mathcal{I},\ \forall \epsilon \in \{-1, +1\}, \quad \partial_{r_{\epsilon k}} \hat{\jmath}_e(\boldsymbol{r}_{\mathcal{I}}, \varsigma) = \sum_{i \in \mathcal{I}} \hat{\lambda}^{\mathcal{I}}_{e\leftarrow i}(\varsigma)\, \partial_{r_{\epsilon k}} \hat{\jmath}_i(\boldsymbol{r}_{\mathcal{I}}, \varsigma). \tag{25}$$

The key technical ingredient is the expression, obtained in Ref. [33], of the partial derivative of currents with respect to rates $r_{\epsilon k}$ (with $\epsilon \in \{-1, +1\}$), as proportional to a matrix element of the resolvent of $\mathbf{V}^\top \mathbf{S}$:

$$\partial_{r_{\epsilon k}} \hat{\jmath}_i(\varsigma) = \epsilon\, \varsigma\, \big\langle \mathbf{s}(\epsilon k) \big| \hat{\mathbf{p}}(\varsigma) \big\rangle \big\langle i \big| \big(\varsigma\mathbf{1} + \mathbf{V}^\top \mathbf{S}\big)^{-1} \big| k \big\rangle. \tag{26}$$

Here $\mathbf{s}(+k)$ (resp. $\mathbf{s}(-k)$) is the source (resp. target) of edge $k$, and $\mathbf{S}$ and $\mathbf{V}$ are respectively the stoichiometric and the current matrices of dimensions $X \times E$ and entries[1]

$$\mathbf{S}_{\mathbf{x},e} = \delta_{\mathbf{x},\mathbf{s}(-e)} - \delta_{\mathbf{x},\mathbf{s}(+e)} \tag{27}$$

$$\mathbf{V}_{\mathbf{x},e} = r_{-e}\, \delta_{\mathbf{x},\mathbf{s}(-e)} - r_{+e}\, \delta_{\mathbf{x},\mathbf{s}(+e)}, \tag{28}$$

where in this section we define $X = |\mathscr{X}|$ and $E = |\mathscr{E}|$ for compactness. We also use the bra-ket notation: $|k\rangle$ (resp. $\langle k|$) denotes the canonical column (resp. line) vector along the $k$-th direction.

---

[1] The current writes as $\jmath(t) = -\mathbf{V}^\top \mathbf{p}(t)$ and the rate matrix as $\mathbf{R} = -\mathbf{S}\mathbf{V}^\top$. See Refs [33, 45, 47] for example uses.

The interest of Eq. (26) is that the prefactors of the matrix element of the resolvent often vanish by compensation. And indeed since Perron–Frobenius ensures $\langle \mathbf{s}(\epsilon k) | \hat{\mathbf{p}}(\varsigma) \rangle > 0$, we see from Eq. (25) that we have to prove

$$\exists \hat{\boldsymbol{\lambda}}^{\mathcal{I}}(\varsigma) : \forall k \in \mathcal{I}, \quad \langle e | (\varsigma \mathbf{1} + \mathbf{V}^{\top} \mathbf{S})^{-1} | k \rangle = \sum_{i \in \mathcal{I}} \hat{\lambda}^{\mathcal{I}}_{e \leftarrow i}(\varsigma) \langle i | (\varsigma \mathbf{1} + \mathbf{V}^{\top} \mathbf{S})^{-1} | k \rangle \tag{29}$$

(notice the essential disappearance of the sign $\epsilon$), where $\hat{\boldsymbol{\lambda}}^{\mathcal{I}}(\varsigma)$ has to be independent on the $2n$ rates $\boldsymbol{r}_{\mathcal{I}}$. Now, denoting $\mathbf{M} = \varsigma \mathbf{1} + \mathbf{V}^{\top} \mathbf{S}$ and $(\mathbf{M}^{-1})_{I,J}$ the submatrix of $\mathbf{M}^{-1}$ where only lines and columns of tuples of distinct indices $I$ and $J$ are kept, we see that the condition of Eq. (29) transforms into a matricial equation for the vector $\hat{\boldsymbol{\lambda}}^{\mathcal{I}}(\varsigma)$:

$$\mathbf{M}^{-1}{}^{\top}_{\mathcal{I},e} = \mathbf{M}^{-1}{}^{\top}_{\mathcal{I},\mathcal{I}} \hat{\boldsymbol{\lambda}}^{\mathcal{I}}(\varsigma). \tag{30}$$

We now prove that this equation can be solved by inverting the $n \times n$ matrix $\mathbf{M}^{-1}{}^{\top}_{\mathcal{I},\mathcal{I}}$, and that the result does not depend on the rates $\boldsymbol{r}_{\mathcal{I}}$ of the input currents. A central identity to do so is Jacobi's formula, which states that for an invertible matrix $\mathbf{B}$, one has

$$\det (\mathbf{B}^{-1})_{I,J} = \varepsilon_{IJ} \frac{\det \mathbf{B}_{\smallsetminus (J,I)}}{\det \mathbf{B}}, \tag{31}$$

with $\mathbf{B}_{\smallsetminus (J,I)}$ the submatrix of $\mathbf{B}$ where lines $J$ and columns $I$ are removed, and $\varepsilon_{IJ} \in \{-1, +1\}$ is a sign that depends on the tuples $I$ and $J$:

$$\varepsilon_{IJ} = (-1)^{\sum_{i \in I} i + \sum_{j \in J} j} \sigma(I) \sigma(J). \tag{32}$$

Here $\sigma(I)$ is the signature of $I$, seen as the permutation of its elements sorted by increasing order (for instance $I = (1, 4, 2)$ is seen as the permutation of $(1, 2, 4)$ so that $\sigma(I) = -1$). We thus have[2]

$$\det \mathbf{M}^{-1}{}^{\top}_{\mathcal{I},\mathcal{I}} = \frac{\det \mathbf{M}_{\smallsetminus (\mathcal{I},\mathcal{I})}}{\det \mathbf{M}} = \frac{\det \left( \varsigma \mathbf{1} + (\mathbf{V}_{\smallsetminus (\cdot,\mathcal{I})})^{\top} \mathbf{S}_{\smallsetminus (\cdot,\mathcal{I})} \right)}{\det \left( \varsigma \mathbf{1} + \mathbf{V}^{\top} \mathbf{S} \right)} = \varsigma^{-n} \frac{\det \left( \varsigma \mathbf{1} - \mathbf{R}_{\smallsetminus \mathcal{I}} \right)}{\det \left( \varsigma \mathbf{1} - \mathbf{R} \right)}, \tag{33}$$

where $\mathbf{R}_{\smallsetminus \mathcal{I}}$ is the rate matrix of a reduced network where the $n$ input edges $\mathcal{I}$ are removed. The matrix $\mathbf{M}^{-1}{}^{\top}_{\mathcal{I},\mathcal{I}}$ is thus invertible for complex $\varsigma$ not belonging to the spectra of $\mathbf{R}$ or $\mathbf{R}_{\smallsetminus \mathcal{I}}$ and thus, since these matrices are both stochastic, in a finite vicinity of 0 in $\mathbb{C} \smallsetminus \{0\}$ (which will allow us to study the $\varsigma \to 0$ limit safely).

Eq. (30) is a linear equation of the form $\mathbf{y} = \mathbf{B} \mathbf{x}$ with $\mathbf{x}$ and $\mathbf{y}$ two vectors of length $n$ and $\mathbf{B}$ a $n \times n$ invertible matrix. Cramer's rule gives its solution in a determinantal form: the component $i$ of $\mathbf{x}$ writes as $x_i = \det \mathbf{B}_i / \det \mathbf{B}$ where $\mathbf{B}_i$ is obtained replacing column $i$ of $\mathbf{B}$ by the vector $\mathbf{y}$. In the case of Eq. (30), the vector on the l.h.s. is equal to the column labelled by $e$ of the matrix on the r.h.s. Cramer's rule thus offers an explicit expression of the components of the vector $\hat{\boldsymbol{\lambda}}^{\mathcal{I}}(\varsigma)$ as

$$\hat{\lambda}^{\mathcal{I}}_{e \leftarrow i}(\varsigma) = \frac{\det \left[ \mathbf{M}^{-1}{}_{\mathcal{I}|_{i \to e}, \mathcal{I}} \right]}{\det \left[ \mathbf{M}^{-1}{}_{\mathcal{I},\mathcal{I}} \right]} \tag{34}$$

$$= (-1)^{i+e} \sigma(\mathcal{I}) \sigma(\mathcal{I}|_{i \to e}) \frac{\det \left( \varsigma \mathbf{1} + \mathbf{V}^{\top} \mathbf{S} \right)_{\smallsetminus (\mathcal{I}, \mathcal{I}|_{i \to e})}}{\det \left( \varsigma \mathbf{1} + \mathbf{V}^{\top} \mathbf{S} \right)_{\smallsetminus (\mathcal{I}, \mathcal{I})}}, \tag{35}$$

---

[2]We use Sylvester's determinant theorem to write: $\det \left( \varsigma \mathbf{1} + (\mathbf{V}_{\smallsetminus (\cdot,\mathcal{I})})^{\top} \mathbf{S}_{\smallsetminus (\cdot,\mathcal{I})} \right) = \varsigma^{E-n-X} \det \left( \varsigma \mathbf{1} - \mathbf{R}_{\smallsetminus \mathcal{I}} \right)$, and $\det \mathbf{M} = \varsigma^{E-X} \det(\varsigma \mathbf{1} - \mathbf{R})$.

where $\mathcal{I}|_{i\mapsto e}$ denotes the tuple obtained from $\mathcal{I}$ by replacing label $i$ by $e$. For the second line we make use of Jacobi's identity, Eq. (31). From Eq. (35), we see explicitly that the susceptibility $\hat{\lambda}^{\mathcal{I}}_{e\leftarrow i}(\varsigma)$ is independent of the rates of the input edges $\mathcal{I}$, since $\mathbf{V}^\top\mathbf{S}$ depends on the two rates $\boldsymbol{r}_i$ only through its line $i$. (This is not obvious to read from Eq. (34) since the matrices involved on its r.h.s. depend explicitly on the rates $\boldsymbol{r}_i$.) This ends the proof of the Laplace-domain multilinearity, Eq. (24).

## 4.3 Taking the limit $\varsigma \to 0$ and mapping to the one-input-current case

Using Sylvester's determinant theorem, the denominator of the expression of Eq. (35) of the Laplace-domain susceptibility $\hat{\lambda}^{\mathcal{I}}_{e\leftarrow i}(\varsigma)$ rewrites as

$$\det\left(\varsigma\mathbf{1}+\mathbf{V}^\top\mathbf{S}\right)_{\smallsetminus(\mathcal{I},\mathcal{I})} \quad = \varsigma^{E-n-X}\det(\varsigma\mathbf{1}-\mathbf{R}_{\smallsetminus\mathcal{I}}) \tag{36}$$

$$\overset{\varsigma\to 0}{\sim} \varsigma^{E-n-X+1}\,\mathrm{tr\,adj}\left(-\mathbf{R}_{\smallsetminus\mathcal{I}}\right) \tag{37}$$

where the trace of the adjugate of $-\mathbf{R}_{\smallsetminus\mathcal{I}}$ can be expressed as a sum of weights of rooted spanning trees of $\mathscr{G}_{\smallsetminus\mathcal{I}}$, the Markov chain graph where edges from $\mathcal{I}$ are removed (this is an example of relation between determinants and spanning tree expansions [48], and see e.g. Refs. [33, 45, 49] for recent applications in the field of Markov chains).

   Understanding the $\varsigma \to 0$ behavior of the numerator of Eq. (35) proves to be a little trickier. Consider the graph $\tilde{\mathscr{G}}$ obtained from $\mathscr{G}$ by removing the edges in $\mathcal{I} \smallsetminus \{i\}$, where we fix for a moment an input edge $i \in \mathcal{I}$. This defines an auxiliary one-input-current problem, with input edge $i$ and output edge $e$ in the reduced graph $\tilde{\mathscr{G}}$. The graph $\tilde{\mathscr{G}}$ has $X$ vertices and $\tilde{E} = E - n + 1$ edges, and is connected (since removing $\mathcal{I}$ does not disconnect $\mathscr{G}$). As a special case of Eq. (35), the corresponding Laplace-domain current-current susceptibility, that we denote $\tilde{\lambda}^i_{e\leftarrow i}(\varsigma)$, is expressed as

$$\tilde{\lambda}^i_{e\leftarrow i}(\varsigma) = (-1)^{\tilde{i}+\tilde{e}}\frac{\det\left(\varsigma\mathbf{1}_{\tilde{E}}+\tilde{\mathbf{V}}^\top\tilde{\mathbf{S}}\right)_{\smallsetminus(\tilde{i},\tilde{e})}}{\det\left(\varsigma\mathbf{1}_{\tilde{E}}+\tilde{\mathbf{V}}^\top\tilde{\mathbf{S}}\right)_{\smallsetminus(\tilde{i},\tilde{i})}} \tag{38}$$

where $\tilde{\mathbf{S}}$ is the stoichiometric matrix of $\tilde{\mathscr{G}}$ and $\tilde{\mathbf{V}}$ its corresponding weighted version, defined similarly to Eqs. (27)-(28) (also, to facilitate the reading until the end of this section, we make explicit the dimension of identity matrices as indices). The integers $\tilde{i}$ and $\tilde{e}$ are the indices of input and output edges in the reduced set of edges of $\tilde{\mathscr{G}}$, so that[3] $(-1)^{\tilde{i}+\tilde{e}} = (-1)^{i+e}\,\sigma(\mathcal{I})\,\sigma(\mathcal{I}|_{i\mapsto e})$. Using Sylvester's determinant theorem as previously, we have:

$$\det\left(\varsigma\mathbf{1}_{\tilde{E}}+\tilde{\mathbf{V}}^\top\tilde{\mathbf{S}}\right)_{\smallsetminus(\tilde{i},\tilde{i})} = \varsigma^{\tilde{E}-1-X}\det(\varsigma\mathbf{1}_X-\tilde{\mathbf{R}}_{\smallsetminus\tilde{i}}) \overset{(36)}{=} \det\left(\varsigma\mathbf{1}_E+\mathbf{V}^\top\mathbf{S}\right)_{\smallsetminus(\mathcal{I},\mathcal{I})} \tag{39}$$

where we used $\tilde{\mathbf{R}}_{\smallsetminus\tilde{i}} = \mathbf{R}_{\smallsetminus\mathcal{I}}$. Also, by direct inspection of definitions, we have:

$$\left(\varsigma\mathbf{1}_{\tilde{E}}+\tilde{\mathbf{V}}^\top\tilde{\mathbf{S}}\right)_{\smallsetminus(\tilde{i},\tilde{e})} = \left(\varsigma\mathbf{1}_E+\mathbf{V}^\top\mathbf{S}\right)_{\smallsetminus(\mathcal{I},\mathcal{I}|_{i\mapsto e})}. \tag{40}$$

Collecting these results, we finally obtain

$$\hat{\lambda}^{\mathcal{I}}_{e\leftarrow i}(\varsigma) = \tilde{\lambda}^i_{e\leftarrow i}(\varsigma). \tag{41}$$

---

[3] From the definition of $\tilde{i}$ and $\tilde{e}$, we see that $(-1)^{\tilde{i}-\tilde{e}} = (-1)^{i-e}(-1)^\ell$ where $\ell$ is the cardinality of $\mathcal{J} = \mathcal{I}\cap]i,e[$, the set of elements of $\mathcal{I}$ strictly comprised between the two integers $i$ and $e$. Also, the composition of $\mathcal{I}|_{i\mapsto e}$ and of the inverse of $\mathcal{I}$ (here, as discussed after Eq. (32), a sequence of integers is seen as the permutation of its elements sorted by increasing order) is a cycle permutation of length $\ell+1$, because it shifts $\{e\}\cup\mathcal{J}$ to $\mathcal{J}\cup\{i\}$. Its signature $\sigma(\mathcal{I}^{-1}\circ\mathcal{I}|_{i\mapsto e}) = \sigma(\mathcal{I})\sigma(\mathcal{I}|_{i\mapsto e})$ is thus equal to $(-1)^\ell$, which proves the announced result.

In other words, the susceptibility of the output current $e$ in the input current $i \in \mathcal{I}$ in the graph $\mathcal{G}$ is equal to the (one-input-current) susceptibility for the same input and output edges but in the reduced graph $\tilde{\mathcal{G}}$ where input edges of $\mathcal{I}$ other than $i$ are removed. Eq. (41) thus provides a interesting physical representation of the susceptibility in the Laplace-space mutual linearity of Eq. (24): the susceptibility $\hat{\lambda}^{\mathcal{I}}_{e \leftarrow i}(\varsigma)$ of the full problem can be computed from that of a simpler problem (one input and one output current, in a reduced graph).

Coming back to the $\varsigma \to 0$ asymptotics, Eq. (41) implies that $\lambda^{\mathcal{I}}_{e \leftarrow i} = \lim_{\varsigma \to 0} \hat{\lambda}^{\mathcal{I}}_{e \leftarrow i}(\varsigma)$ exists and can be expressed, following Ref. [33], as a sum of weights of trees constructed from the reduced graph $\tilde{\mathcal{G}}$. This ends the Laplace-space proof of the stationary MML [Eq. (7)], and since Eq. (41) extends to the $\varsigma \to 0$ limit, also provides a representation of the stationary susceptibility $\lambda^{\mathcal{I}}_{e \leftarrow i}$ as the one, $\tilde{\lambda}^{i}_{e \leftarrow i}$, of a simpler problem.

## 5 Further developments

### 5.1 Recovering Kirchhoff's Current Law

MML in Eq. (7) generalizes KCL in the sense that it extends the validity of Eq. (6) to subsets of currents that do not constitute a fundamental set, cf. Fig. 2. Importantly, we now show that Eq. (7) yields back Eq. (6) when the input set $\mathcal{I}$ is fundamental as a result of the terms $J^{\smallsetminus \mathcal{I}}_e$ vanishing and the terms $\lambda^{\mathcal{I}}_{e \leftarrow i}$ reducing to 0 or $\pm 1$.

Let $\mathcal{I}$ be a fundamental, and therefore also admissible, set. It is immediate that $J^{\smallsetminus \mathcal{I}}_e = 0$ for all edges $e$ since there are no stationary currents on a tree.

Regarding the susceptibility $\lambda^{\mathcal{I}}_{e \leftarrow i}$ of a current over $e$ with respect to $i \in \mathcal{I}$, we have shown in Sec. 4.3 that it is equivalent to the susceptibility $e \leftarrow i$ when $i$ is a single input current in the reduced graph obtain from $\mathcal{G}$ by removing the edges $\mathcal{I} \smallsetminus \{i\}$. Then, using the expression derived in Ref. [33] of such a single-input-current susceptibility in terms of spanning tree polynomials, we arrive at:

$$\lambda^{\mathcal{I}}_{e \leftarrow i} = r_{\mathbf{s}(-e) \leftarrow \mathbf{s}(+e)} \frac{\tau^{\nearrow s(+e) \to s(+i)}_{\searrow \mathcal{I}, e} - \tau^{\nearrow s(+e) \to s(-i)}_{\searrow \mathcal{I}, e}}{\tau_{\searrow \mathcal{I}}} - r_{\mathbf{s}(+e) \leftarrow \mathbf{s}(-e)} \frac{\tau^{\nearrow s(-e) \to s(+i)}_{\searrow \mathcal{I}, e} - \tau^{\nearrow s(-e) \to s(-i)}_{\searrow \mathcal{I}, e}}{\tau_{\searrow \mathcal{I}}},$$
(42)

where

$$\tau_{\searrow \mathcal{I}} := \sum_{\mathbf{x}} \sum_{\mathcal{T}_{\mathbf{x}} \subseteq \tilde{\mathcal{G}}} \omega(\mathcal{T}_{\mathbf{x}}) \tag{43}$$

is the sum over all vertices $\mathbf{x}$ and over all rooted spanning trees $\mathcal{T}_{\mathbf{x}}$ in $\tilde{\mathcal{G}} := \mathcal{G} \smallsetminus \mathcal{I}$, of the product of the transition rates $\omega(\mathcal{T}_{\mathbf{x}})$ belonging to $\mathcal{T}_{\mathbf{x}}$; also,

$$\tau^{\nearrow \mathbf{x} \to \mathbf{y}}_{\searrow \mathcal{I}, e} := \sum_{\mathbf{x}} \sum_{\mathcal{T}_{\mathbf{x}} \subseteq \mathcal{G}' : \{\mathbf{x} \to \mathbf{y}\} \subseteq \mathcal{T}_{\mathbf{x}}} \omega(\mathcal{T}_{\mathbf{x}}) \tag{44}$$

is a similar sum of polynomials for the graph $\mathcal{G}' := (\tilde{\mathcal{G}} \cup \{\mathbf{x} \to \mathbf{y}\}) \smallsetminus \{e\}$ where an auxiliary edge $\mathbf{x} \to \mathbf{y}$ of rate 1 is introduced while all edges in $\mathcal{I}$ and edge $e$ are removed, and only rooted spanning trees that contain this auxiliary edge are considered. We will use Eq. (42) to explicitly obtain the susceptibilities in this case.

Since $\mathcal{I}$ constitutes a fundamental set, the first step of removing $\mathcal{I} \smallsetminus \{i\}$ leads to a single cycle $\mathcal{C}$ and possibly branching leaves. Secondly, removing $i$ results in a tree $\mathcal{T} = \mathcal{E} \smallsetminus \mathcal{I}$. Lastly, two disconnected trees sprout from the removal of $e$, $\mathcal{T}^+$ and $\mathcal{T}^-$. The former contains $\mathbf{s}(+e) \subseteq \mathcal{T}^+$ while $\mathbf{s}(-e) \subseteq \mathcal{T}^-$, see Fig. 4.

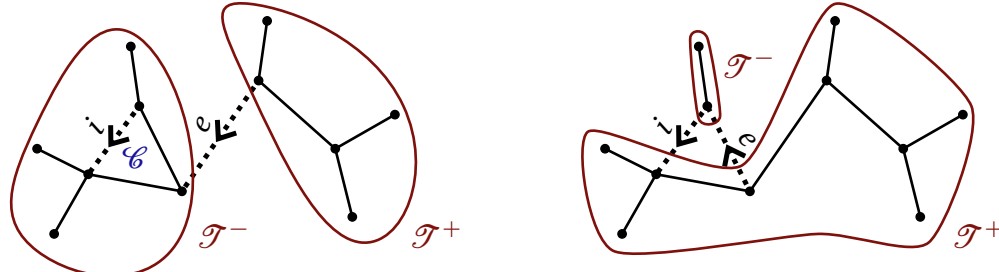

Figure 4: *Resulting graph after removal of $\mathcal{I}$ and $e$* Left: $e$ and $i$ do not belong to the same cycle $\mathscr{C}$, notice that $i$ is fully contained by $\mathscr{T}^-$. Right: Since $e$ and $i$ belong to the same cycle, $i$ is in between both trees $\mathscr{T}^{\pm}$.

We first consider the case $e \notin \mathscr{C}$ (left panel of Fig. 4), in which case $\mathtt{s}(\pm i)$ belong to one of the two trees. If $i \subseteq \mathscr{T}^-$,

$$\tau_{\searrow \mathcal{I},e}^{\nearrow \mathtt{s}(+e)\to \mathtt{s}(+i)} = \tau_{\searrow \mathcal{I},e}^{\nearrow \mathtt{s}(+e)\to \mathtt{s}(-i)} = \omega\big(\mathscr{T}^+_{\mathtt{s}(+e)}\big) \sum_{\mathtt{x}\in \mathscr{T}^-} \omega\big(\mathscr{T}^-_{\mathtt{x}}\big) \tag{45}$$

because the added edges $\mathtt{s}(+e) \to \mathtt{s}(\pm i)$ act as bridges between the tree $\mathscr{T}^+$ and $\mathscr{T}^-$, while

$$\tau_{\searrow e,\mathcal{I}}^{\nearrow \mathtt{s}(-e)\to \mathtt{s}(+i)} = \tau_{\searrow e,\mathcal{I}}^{\nearrow \mathtt{s}(-e)\to \mathtt{s}(-i)} = 0 \tag{46}$$

because $\mathtt{s}(\pm i)$ and $\mathtt{s}(-e)$ belong all to $\mathscr{T}^-$ and the resulting graph is disconnected (no edge connects $\mathscr{T}^+$ and $\mathscr{T}^-$) and therefore has no spanning trees. Consequently, $\lambda^{\mathcal{I}}_{e\leftarrow i} = 0$, and similarly for $i \subseteq \mathscr{T}^+$.

The second case, where $e \in \mathscr{C}$, results in the source and target of $+i$ belonging to distinct trees. If $\mathtt{s}(+i) \subseteq \mathscr{T}^-$,

$$\tau_{\searrow \mathcal{I},e}^{\nearrow \mathtt{s}(+e)\to \mathtt{s}(+i)} = \omega\big(\mathscr{T}^+_{\mathtt{s}(+e)}\big) \sum_{\mathtt{x}\in \mathscr{T}^-} \omega\big(\mathscr{T}^-_{\mathtt{x}}\big),$$
$$\tau_{\searrow \mathcal{I},e}^{\nearrow \mathtt{s}(-e)\to \mathtt{s}(-i)} = \omega\big(\mathscr{T}^-_{\mathtt{s}(-e)}\big) \sum_{\mathtt{x}\in \mathscr{T}^+} \omega\big(\mathscr{T}^+_{\mathtt{x}}\big), \tag{47}$$

and

$$\tau_{\searrow e,\mathcal{I}}^{\nearrow \mathtt{s}(+e)\to \mathtt{s}(-i)} = \tau_{\searrow e,\mathcal{I}}^{\nearrow \mathtt{s}(-e)\to \mathtt{s}(+i)} = 0. \tag{48}$$

Plugging these values in Eq. (42), we identify the numerator as equivalent to $\tau_{\searrow \mathcal{I}}$ and therefore $\lambda^{\mathcal{I}}_{e\leftarrow i} = 1$. Finally, if $\mathtt{s}(+i) \subseteq \mathscr{T}^+$, the same procedure leads to $\lambda^{\mathcal{I}}_{e\leftarrow i} = -1$.

In summary, for an input set that is also fundamental, any output current can be written as a linear combination of input currents with coefficient 0 when $e$ does not lie in the same cycle of $i$ in the graph $\mathscr{E} \smallsetminus (\mathcal{I} \smallsetminus \{i\})$, and $\pm 1$ otherwise, with sign depending on the convention directions, which is Eq. (6).

## 5.2 Connection to equilibrium fluctuations

Here we show that when currents are controlled starting from an equilibrium state, the susceptibility can be interpreted in terms of currents' covariances at equilibrium. This brides with and extends known results in so-called linear regime theory, but in a fully nonequilibrium context.

First we need this:

$$J_e = \lim_{t \to \infty} \frac{\langle J_e(t) \rangle}{t} \tag{49}$$

$$c_{ee'} = \lim_{t \to \infty} \frac{\langle J_e(t) J_{e'}(t) \rangle_c}{t} \tag{50}$$

$$= \lim_{t \to \infty} \frac{\langle J_e(t) J_{e'}(t) \rangle - \langle J_e(t) \rangle \langle J_{e'}(t) \rangle}{t} \tag{51}$$

where $J_e(t)$ are the cumulated currents up to time $t$ along the stochastic process and $\langle \cdot \rangle$ denote sample averages. When the sample mean is over an equilibrium process we have $J_e = 0$ and denote $c_{ee'} = c_{ee'}^{eq}$. Notice that by definition covariances are symmetric: $c_{ee'} = c_{e'e}$.

The main result of this section is that, starting from equilibrium (where $J_e^{eq} = 0$ for all $e$, thus the rates satisfy detailed balance) and perturbing the rates $\boldsymbol{r}_i$ of an input current arbitrarily far from equilibrium, the following linear response relation among currents holds

$$J_e = \frac{c_{ei}^{eq}}{c_{ii}^{eq}} J_i \,, \tag{52}$$

where $c_{e,i}^{eq}$ are the currents' steady covariances at equilibrium. Therefore, properties of response arbitrarily far from equilibrium can be inferred from equilibrium fluctuations.

The proof of the result is straightforward given the results of Refs. [33, 38] and linear response theory (see e.g. [50] for a review and [51] for closely related results). For a moment, let us parametrize the rates according to Eqs. (8)-(9), where we fix the potentials $u_{\mathbf{x} \in \mathscr{X}}$ and take the cycle affinities $\boldsymbol{a} = (a_\alpha)$ (where $\alpha$ runs through chords) as control parameters to drive the dynamics out of the equilibrium reference state $\boldsymbol{a} = \boldsymbol{0}$. The following fluctuation-dissipation relation holds (coming from linear response theory, see App. D for a full derivation):

$$\left. \frac{\partial J_e}{\partial a_\alpha} \right|_{\text{eq.}} = \frac{1}{2} c_{e\alpha}^{eq} \,. \tag{53}$$

Consider first the case of one admissible input current $i$ (that can thus be taken as a chord). From the results of Ref. [33], there exist two constants $J_e^{\searrow i}$ and $\lambda_{e \leftarrow i}^i$ such that for all values of the rates $\boldsymbol{r}_i$, we have $J_e(\boldsymbol{r}_i) = J_e^{\searrow i} + \lambda_{e \leftarrow i}^i J_i(\boldsymbol{r}_i)$. If we parametrize the rates by fixing the potentials $u_{\mathbf{x} \in \mathscr{X}}$ and affinities $a_{\alpha \neq i}$, while controlling the currents through $a_i$, this also means that we have (with the same constants)

$$J_e(a_i) = J_e^{\searrow i} + \lambda_{e \leftarrow i}^i J_i(a_i). \tag{54}$$

For rates satisfying detailed balance ($\boldsymbol{a} = \boldsymbol{0}$) we have $J_e^{\searrow i}|_{eq} = 0$. Taking the derivative of Eq. (54) with respect to $a_i$, using Eq. (53) and rearranging terms we obtain $\lambda_{e \leftarrow i}^i{}^{eq} = c_{ei}^{eq}/c_{ii}^{eq}$ and thus Eq. (52).

The generalization to an arbitrary number of admissible input rates being modified is straightforward. Multilinearity reads

$$J_e(\boldsymbol{r}_{\mathcal{I}}) = J_e^{\searrow \mathcal{I}} + \sum_{i \in \mathcal{I}} \lambda_{e \leftarrow i}^{\mathcal{I}} J_i(\boldsymbol{r}_{\mathcal{I}}) \,. \tag{55}$$

We parametrize the rates with a potential $u_{\mathbf{x} \in \mathscr{X}}$ and cycle affinities $\boldsymbol{a}$, and we consider equilibrium reference rates ($\boldsymbol{a} = \boldsymbol{0}$). We define the vectors $\boldsymbol{\lambda}_e^{\mathcal{I}} = (\lambda_{e \leftarrow i}^{\mathcal{I}}{}^{eq})_{i \in \mathcal{I}}$, $\boldsymbol{c}_e^{\mathcal{I}} = (c_{e,i}^{eq})_{i \in \mathcal{I}}$, $\boldsymbol{J}_{\mathcal{I}} = (J_i)_{i \in \mathcal{I}}$, $\boldsymbol{a}_{\mathcal{I}} = (a_i)_{i \in \mathcal{I}}$, and the matrix $\mathbf{C}_{\mathcal{I}} = (c_{k,i}^{eq})_{k,i \in \mathcal{I}}$. Controlling currents through $\boldsymbol{a}_{\mathcal{I}}$, Eq. (55) implies[4]

$$J_e(\boldsymbol{a}_{\mathcal{I}}) = \sum_{i \in \mathcal{I}} \lambda_{e \leftarrow i}^{\mathcal{I}}{}^{eq} J_i(\boldsymbol{a}_{\mathcal{I}}). \tag{56}$$

---

[4]In this expression, explicitly, $\lambda_{e \leftarrow i}^{\mathcal{I}}{}^{eq}$ is given by $\lambda_{e \leftarrow i}^{\mathcal{I}}$ evaluated for cycle affinities $a_\alpha = 0$, $\forall \alpha \notin \mathcal{I}$.

Since $\mathcal{I}$ is admissible, it can be taken as a subset of chords, that we fix and this allows one to use Eq. (53) in this expression, which yields $\boldsymbol{c}_e^{\mathcal{I}} = \mathbf{C}_{\mathcal{I}} \boldsymbol{\lambda}_e^{\mathcal{I}}$. Now, we remark that $\mathbf{C}_{\mathcal{I}}$ is a principal submatrix of the full (i.e. all-chord) response matrix between cycle affinities and chord currents, which is symmetric definite positive (see e.g. [38]). Hence $\mathbf{C}_{\mathcal{I}}$ has the same property and is invertible. This allows one to write $\boldsymbol{\lambda}_e^{\mathcal{I}} = \mathbf{C}_{\mathcal{I}}^{-1} \boldsymbol{c}_e^{\mathcal{I}}$ and we obtain finally

$$J_e(\boldsymbol{r}_{\mathcal{I}}) = \boldsymbol{c}_e^{\mathcal{I}} \cdot \mathbf{C}_{\mathcal{I}}^{-1} \boldsymbol{J}_{\mathcal{I}}(\boldsymbol{r}_{\mathcal{I}}) , \tag{57}$$

valid when controlling input currents starting from an equilibrium reference point.

Its physical relevance of our result is that it is phenomenological, in that the currents' response coefficients do not have to be obtained analytically, but can be sampled from a realization of the process.

# 6 Conclusion and Perspectives

The question of how stationary currents on a Markov process reorganize when tuning transition rates is relevant across a variety of systems, from chemical networks, resistor networks, mechanical systems, and metabolic pathways. In this work, we have uncovered a mutual linear-affine relation between stationary currents when controlling multiple transition rates. Borrowing approaches and terminology from graph theory, we have considered a subset of chord currents as input currents, whose rates can be arbitrarily and independently manipulated. In this context, the Kirchhoff Current Law is a tenet of the theory of Markov chains, which states that a complete set of chord currents fully specifies the currents at stationarity. Our result generalizes Kirchhoff Current Law by drawing a (linear-affine) relation between any stationary current and an incomplete set of chords. As such, it also generalized the recent result put forward by us in Ref. [33] for the case of a single input current.

Our result is remarkably general, being valid for arbitrary network geometries and arbitrarily large changes of rates. Yet, we envisage a few limitations which call for future work. Extending our results to the framework of open (driven) systems is an interesting direction. In the case of a single input current we have proved that the linearity extends to the case of open systems despite the fact that global conservation of probability breaks down [33]. This suggests that such a generalization is viable also for multiple input currents and could lead to interesting applications in the field of adaptive physical networks such as flow, elastic and resistor networks [52–54]. A second, more far-fetched generalization is to incorporate non-linear interacting networks. This is the case of biochemical and metabolic networks, where interactions involve more than two chemicals (for example due to autocatalytic motifs [55]) and therefore maps to hypergraphs instead of Markov chains. Despite the apparent profound difference, recent work has uncovered similarities between noninteracting and interacting networks [38], which encourage one to pursue this path.

We conclude with a final, practical remark. Despite the result presented in this work representing a generalization from single to multiple input currents, it does not include the case of macroscopic input currents that are a linear combination of multiple edge currents. For example, suppose the case where two edge currents $J_1$ and $J_2$ are not independently measurable, but $J = J_1 + J_2$ is measurable. Our result states that

$$J_e = \lambda_e^{\backslash\{1,2\}} + \lambda_{e,1}^{\{1,2\}} J_1 + \lambda_{e,2}^{\{1,2\}} J_2 \tag{58}$$

$$= \lambda_e^{\backslash\{1,2\}} + (\lambda_{e,1}^{\{1,2\}} - \lambda_{e,2}^{\{1,2\}}) J_1 + \lambda_{e,2}^{\{1,2\}} J \tag{59}$$

and we immediately observe that we cannot get rid of the unmeasurable current $J_1$ unless $\lambda_{e,1}^{\{1,2\}} = \lambda_{e,2}^{\{1,2\}}$. As pointed out in Ref. [33], however, the result easily extends to the case where the macroscopic output currents are linear combinations of edge currents.

# Acknowledgements

We warmly thank Faezeh Khodabandehlou and Christian Maes for discussions. MP is grateful to Villa ai Colli Research Institute for hospitality.

**Funding information**     PH was supported by the project INTER/FNRS/20/15074473 funded by F.R.S.-FNRS (Belgium) and FNR (Luxembourg). SDC and VL acknowledge support from IXXI, CNRS MITI, CNRS Tremplin and the ANR-18-CE30-0028-01 grant LABS.

# Appendices

# A    Construction of potentials and affinities from transition rates

Given a spanning tree $\mathcal{T}$ with an arbitrarily fixed root $\mathrm{x}_0$, we define for $\mathrm{x} \leftarrow \mathrm{y} \in \overline{\mathcal{T}}$ the local affinity $A_{\mathrm{x} \leftarrow \mathrm{y}} := \log \frac{r_{\mathrm{x} \leftarrow \mathrm{y}}}{r_{\mathrm{y} \leftarrow \mathrm{x}}}$. For any vertex $\mathrm{z} \in \mathcal{X}$, there is a unique path in $\mathcal{T}$ going from $\mathrm{z}$ to $\mathrm{x}_0$, that we denote $\mathscr{P}_{\mathrm{z}} = (\mathrm{x}_0 \leftarrow \mathrm{x}_1, \mathrm{x}_1 \leftarrow \mathrm{x}_2, \dots, \mathrm{x}_p \leftarrow \mathrm{z})$. The potential in $\mathrm{z}$ is defined as the "integral" of the local affinity along this path

$$u_{\mathrm{z}} = \sum_{\mathrm{x} \leftarrow \mathrm{y} \in \mathscr{P}_{\mathrm{z}}} A_{\mathrm{x} \leftarrow \mathrm{y}} \,. \tag{A.1}$$

This implies that the local affinity is the (discrete) "gradient" of the potential: $u_{\mathrm{y}} - u_{\mathrm{x}} = A_{\mathrm{x} \leftarrow \mathrm{y}}$ for $\mathrm{x} \leftarrow \mathrm{y} \in \overline{\mathcal{T}}$. This relation ceases to be true in general on a chord $\mathrm{x} \leftarrow \mathrm{y} = \alpha \in \mathscr{E} \setminus \mathcal{T}$ (unless the Wegscheider condition is satisfied along its associated cycle $\mathscr{C}_\alpha$). The so-called cycle affinity $a_\alpha := \log \frac{r_{\mathrm{x} \leftarrow \mathrm{y}}}{r_{\mathrm{y} \leftarrow \mathrm{x}}} - u_{\mathrm{y}} + u_{\mathrm{x}}$ quantifies precisely how it is broken. The potential and cycle affinity so defined verify the relations of Eqs. (8)-(9) of the main text. Note that the potential depends only on the spanning-tree rates $(r_e)_{e \in \mathcal{T}}$, while the cycle affinity depends on these rates and on the rates $r_\alpha$ of the chord (more specifically, it depends on the cycle rates $(r_e)_{e \in \mathscr{C}_\alpha}$).

# B    Linear independence of admissible currents

An admissible set $\mathcal{I}$ of edges can be represented as a subset of a fundamental set of chords associated to a spanning tree $\mathcal{T}$ (i.e. $\mathcal{I} \subseteq \mathscr{E} \setminus \mathcal{T}$). It is thus natural to expect that their corresponding currents $(J_i)_{i \in \mathcal{I}}$ are linearly independent when varying the transition rates of the Markov chain. The following lemma makes this statement precise.

**Independence lemma**: Fix the rates $r_{\mathscr{E} \setminus \mathcal{I}}$ of the transitions not in the admissible set. Then, denoting by $J_i(r_{\mathcal{I}})$ the stationary current seen as a function of the remaining rates $r_{\mathcal{I}}$,

$$\exists \lambda^0, (\lambda_i)_{i \in \mathcal{I}} : \forall r_{\mathcal{I}}, \; \lambda^0 + \sum_{i \in \mathcal{I}} \lambda_i J_i(r_{\mathcal{I}}) = 0 \qquad \Rightarrow \qquad \lambda^0 = 0 \text{ and } \forall i \in \mathcal{I}, \lambda_i = 0 \,.$$

*Proof* — Sending the rates $r_{\mathcal{I}}$ to 0, the stationary currents $(J_i)_{i \in \mathcal{I}}$ also go to 0, which implies $\lambda^0 = 0$. Now, fix $i \in \mathcal{I}$. This chord is associated to a cycle $\mathscr{C}_i$ which does not involve the other edges in $\mathcal{I}$. Sending $(r_\iota)_{\iota \in \mathcal{I} \setminus \{i\}}$ to 0 sends $(J_\iota)_{\iota \in \mathcal{I} \setminus \{i\}}$ to 0, and by hypothesis, we are left with

$$\forall r_i, \; \lambda_i J_i(r_{\mathcal{I}}) \big|_{r_{\iota \neq i} = 0} = 0 \,. \tag{B.1}$$

For any choice of $r_{\mathcal{E} \smallsetminus \mathcal{I}}$, one can find two rates $r_i$ such that the Wegscheider condition along cycle $\mathscr{C}_i$ is not satisfied. For such a choice, one has $J_i \neq 0$. This implies that $\lambda_i = 0$, and since this reasoning can be done for every $i \in \mathcal{I}$, this ends the proof. $\square$

*Remark* — This independence lemma implies that the linear-affine decomposition of Eq. (7) of a current $J_e$ on a set of admissible currents $J_{\mathcal{I}}$ with coefficients depending on the rates $r_{\mathcal{E} \smallsetminus \mathcal{I}}$ is unique, and this for any value of $r_{\mathcal{E} \smallsetminus \mathcal{I}}$.

## C  Proof that $1 - \lambda^{\mathcal{I}_1}_{e_2 \leftarrow e_1}(r_{\mathcal{E} \smallsetminus \mathcal{I}_1}) \lambda^{\mathcal{I}_2}_{e_1 \leftarrow e_2}(r_{\mathcal{E} \smallsetminus \mathcal{I}_2})$ cannot cancel

We first rewrite Eqs. (21)-(22) as

$$J_{e_1} = J^{\searrow \mathcal{I}_2}_{e_1} + \lambda^{\mathcal{I}_2}_{e_1 \leftarrow e_2} J_{e_2} + \sum_{i \in \mathcal{I}} \lambda^{\mathcal{I}_2}_{e_1 \leftarrow i} J_i \qquad (C.1)$$

$$J_{e_2} = J^{\searrow \mathcal{I}_1}_{e_2} + \lambda^{\mathcal{I}_1}_{e_2 \leftarrow e_1} J_{e_1} + \sum_{i \in \mathcal{I}} \lambda^{\mathcal{I}_1}_{e_2 \leftarrow i} J_i \,, \qquad (C.2)$$

where in this Appendix, for short, we omit the arguments of functions, i.e. : $J^{\searrow \mathcal{I}_1}_{e_2} = J^{\searrow \mathcal{I}_1}_{e_2}(r_{\mathcal{E} \smallsetminus \mathcal{I}_1})$, $J^{\searrow \mathcal{I}_2}_{e_1} = J^{\searrow \mathcal{I}_2}_{e_1}(r_{\mathcal{E} \smallsetminus \mathcal{I}_2})$, $\lambda^{\mathcal{I}_1}_{e_2 \leftarrow i} = \lambda^{\mathcal{I}_1}_{e_2 \leftarrow i}(r_{\mathcal{E} \smallsetminus \mathcal{I}_1})$, and $\lambda^{\mathcal{I}_2}_{e_1 \leftarrow i} = \lambda^{\mathcal{I}_2}_{e_1 \leftarrow i}(r_{\mathcal{E} \smallsetminus \mathcal{I}_2})$. Equivalently, one has:

$$\mathbf{M} \begin{pmatrix} J_{e_1} \\ J_{e_2} \end{pmatrix} = \boldsymbol{b} \quad \text{with} \quad \mathbf{M} = \begin{pmatrix} 1 & -\lambda^{\mathcal{I}_2}_{e_1 \leftarrow e_2} \\ -\lambda^{\mathcal{I}_1}_{e_2 \leftarrow e_1} & 1 \end{pmatrix} \quad \text{and} \quad \boldsymbol{b} = \begin{pmatrix} J^{\searrow \mathcal{I}_2}_{e_1} + \sum_{i \in \mathcal{I}} \lambda^{\mathcal{I}_2}_{e_1 \leftarrow i} J_i \\ J^{\searrow \mathcal{I}_1}_{e_2} + \sum_{i \in \mathcal{I}} \lambda^{\mathcal{I}_1}_{e_2 \leftarrow i} J_i \end{pmatrix}, \quad (C.3)$$

which we read as an equation for $J_{e_1}$ and $J_{e_2}$. Provided $\mathbf{M}$ is invertible, we obtain the mutual multilinearity needed in the inductive step of Sec. 3.2 (as also directly shown in the text). We now show *ad absurdum* that $\mathbf{M}$ is always invertible.

Let us assume that $\det \mathbf{M} = 1 - \lambda^{\mathcal{I}_1}_{e_2 \leftarrow e_1} \lambda^{\mathcal{I}_2}_{e_1 \leftarrow e_2} = 0$. Then necessarily both $\lambda^{\mathcal{I}_1}_{e_2 \leftarrow e_1}$ and $\lambda^{\mathcal{I}_2}_{e_1 \leftarrow e_2}$ do not cancel and we have $\lambda^{\mathcal{I}_2}_{e_1 \leftarrow e_2} = 1/\lambda^{\mathcal{I}_1}_{e_2 \leftarrow e_1}$. Using this in Eq. (C.1) and multiplying by $\lambda^{\mathcal{I}_1}_{e_2 \leftarrow e_1}$ we arrive at

$$-J_{e_2} = \lambda^{\mathcal{I}_1}_{e_2 \leftarrow e_1} J^{\searrow \mathcal{I}_2}_{e_1} - \lambda^{\mathcal{I}_1}_{e_2 \leftarrow e_1} J_{e_1} + \sum_{i \in \mathcal{I}} \lambda^{\mathcal{I}_1}_{e_2 \leftarrow e_1} \lambda^{\mathcal{I}_2}_{e_1 \leftarrow i} J_i \,. \qquad (C.4)$$

Adding Eq. (C.2), this yields

$$0 = J^{\searrow \mathcal{I}_1}_{e_2} + \lambda^{\mathcal{I}_1}_{e_2 \leftarrow e_1} J^{\searrow \mathcal{I}_2}_{e_1} + \sum_{i \in \mathcal{I}} \left( \lambda^{\mathcal{I}_1}_{e_2 \leftarrow i} + \lambda^{\mathcal{I}_1}_{e_2 \leftarrow e_1} \lambda^{\mathcal{I}_2}_{e_1 \leftarrow i} \right) J_i \,, \qquad (C.5)$$

which is a linear-affine interdependence of the admissible currents $J_{\mathcal{I}}$ with coefficients in $r_{\mathcal{E} \smallsetminus \mathcal{I}}$ (since $(\mathcal{E} \smallsetminus \mathcal{I}_1) \cup (\mathcal{E} \smallsetminus \mathcal{I}_2) = \mathcal{E} \smallsetminus \mathcal{I}$). From the independence lemma of App. B, we thus have

$$J^{\searrow \mathcal{I}_1}_{e_2} + \lambda^{\mathcal{I}_1}_{e_2 \leftarrow e_1} J^{\searrow \mathcal{I}_2}_{e_1} = 0 \qquad (C.6)$$

$$\forall i \in \mathcal{I}, \quad \lambda^{\mathcal{I}_1}_{e_2 \leftarrow i} + \lambda^{\mathcal{I}_1}_{e_2 \leftarrow e_1} \lambda^{\mathcal{I}_2}_{e_1 \leftarrow i} = 0 \,. \qquad (C.7)$$

We now come back to the matrix equation (C.3). By hypothesis, $\mathbf{M}$ is singular; hence, Eq. (C.3) has solutions provided $\boldsymbol{b} \in \text{Im} \, \mathbf{M}$. Since $\text{Im} \, \mathbf{M} = (\text{Ker} \, \mathbf{M}^{\top})^{\perp}$ and $\text{Ker} \, \mathbf{M}^{\top} = \text{Span} \, (\lambda^{\mathcal{I}_1}_{e_2 \leftarrow e_1}, 1)^{\top}$ this condition is equivalent to $\boldsymbol{b} \cdot (\lambda^{\mathcal{I}_1}_{e_2 \leftarrow e_1}, 1) = 0$, but this is precisely guaranteed by Eqs. (C.6)-(C.7). Now, because $\text{Ker} \, \mathbf{M} = \text{Span} \, (1, \lambda^{\mathcal{I}_1}_{e_2 \leftarrow e_1})^{\top}$, this means that Eq. (C.3) has a family of solutions of the form $(J_{e_1}, J_{e_2})^{\top} = \mathbf{M}^{\oplus} \boldsymbol{b} + \gamma (1, \lambda^{\mathcal{I}_1}_{e_2 \leftarrow e_1})^{\top}$ for any real number $\gamma$, where $\mathbf{M}^{\oplus}$ is the Moore–Penrose pseudo-inverse of $\mathbf{M}^{\oplus}$. This is absurd because then we have several distinct linear-affine decompositions of $J_{e_1}$ on the admissible currents $J_{\mathcal{I}}$ with coefficients depending on $r_{\mathcal{E} \smallsetminus \mathcal{I}}$, which is excluded by the independence lemma. $\square$

# D   A fluctuation-dissipation relation

Along each directed edge $i$, we define the time-integrated current $J_i$ of a trajectory of duration $t$ as the number of times transition $+i$ occurs minus the number of times transition $-i$ occurs. This quantity is time-additive and, upon a transition $e \in \overline{\mathscr{E}}$, evolves as

$$J_i \mapsto J_i + \delta_{e,+i} - \delta_{e,-i} \,. \tag{D.1}$$

Its average and second cumulant are related to the stationary mean and covariance of the current as

$$J_i = \lim_{t \to \infty} \frac{1}{t} \langle J_i \rangle \,, \qquad c_{i_1 i_2} = \lim_{t \to \infty} \frac{1}{t} \langle J_{i_1} J_{i_2} \rangle_c \,. \tag{D.2}$$

We define similarly the entropy production $\Sigma$ of a trajectory of duration $t$ as the logarithm of the ratio between the probability of this trajectory and the probability of its time reverse. It is a time-additive observable that increases at every jump: if transition $e \in \overline{\mathscr{E}}$ occurs, one has

$$\Sigma \mapsto \Sigma + \log \frac{r_{+e}}{r_{-e}} \,. \tag{D.3}$$

From the definitions and from the parametrization Eqs. (8)-(9) of the rates (see App. A), it is related to the currents $J_i$'s by

$$\Sigma = \sum_\alpha a_\alpha J_\alpha + \text{b.t.} \tag{D.4}$$

where the sum runs over the chords $\{\alpha\}$ and the boundary term b.t. depends only on the initial and final configurations of the trajectory. We recall that $a_\alpha$ is the cycle affinity associated to chord $\alpha$. We also see from the definition of $\Sigma$, that for any observable $\mathcal{O}$ depending on the trajectory, we have

$$\langle \mathcal{O}(\text{traj}) \rangle = \left\langle \mathcal{O}(\text{traj}^{\text{R}}) \, e^{-\Sigma(\text{traj}) + \text{b.t.}} \right\rangle \,, \tag{D.5}$$

where $\text{traj}^{\text{R}}$ denotes the time reversed of trajectory traj. For instance $J_i(\text{traj}^{\text{R}}) = -J_i(\text{traj})$.

The cumulant generating function of the currents, defined as

$$\psi(\mathbf{s}) = \lim_{t \to \infty} \frac{1}{t} \left\langle e^{\sum_{e \in \mathscr{E}} s_e J_e} \right\rangle \tag{D.6}$$

is such that $J_e = \partial_{s_e} \psi|_{\mathbf{s}=\mathbf{0}}$ and $c_{e_1 e_2} = \partial_{s_{e_1}} \partial_{s_{e_2}} \psi|_{\mathbf{s}=\mathbf{0}}$. From Eqs. (D.4)-(D.5), it satisfies

$$\psi(\mathbf{s}^{\text{cochord}}, \mathbf{s}^{\text{chord}}, \mathbf{a}) = \psi(-\mathbf{s}^{\text{cochord}}, -\mathbf{a} - \mathbf{s}^{\text{chord}}, \mathbf{a}) \,, \tag{D.7}$$

where we split the full vector $\mathbf{s}$ into its chord contribution $\mathbf{s}^{\text{chord}} = (s_\alpha)$ and its cochord one, and $\mathbf{a} = (a_\alpha)$ represent the affinity vector of the chords. Eq. (D.7) constitutes a fluctuation theorem that (slightly) generalizes the one derived in Ref. [51]. Differentiating it w.r.t. $s_e$ and $a_\alpha$, and then sending $\mathbf{s}$ and $\mathbf{a}$ to zero, we arrive at

$$\left. \frac{\partial^2 \psi(\mathbf{s}, \mathbf{a})}{\partial s_e \partial a_\alpha} \right|_{\substack{\mathbf{s}=\mathbf{0} \\ \mathbf{a}=\mathbf{0}}} = \frac{1}{2} \left. \frac{\partial^2 \psi(\mathbf{s}, \mathbf{a})}{\partial s_e \partial s_\alpha} \right|_{\substack{\mathbf{s}=\mathbf{0} \\ \mathbf{a}=\mathbf{0}}} \,. \tag{D.8}$$

From Eq. (D.2), this implies that, at equilibrium ($\mathbf{a} = 0$, denoted by a superscript 'eq'), the response of any stationary current $J_e$ to a change in the affinity $a_\alpha$ of a chord is proportional to a current covariance:

$$\frac{\partial J_e}{\partial a_\alpha}^{\text{eq}} = \frac{1}{2} c_{e\alpha}^{\text{eq}} \,. \tag{D.9}$$

This constitutes a form of fluctuation-dissipation relation at equilibrium. Because input currents can be taken as a subset of chords, this proves Eq. (53) of the main text.

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
