# Peer review of "Mutual Multilinearity of Nonequilibrium Network Currents"

_SciPost Physics_

## Round 1 · Referee Report · Anonymous (Referee 1) · 2025-5-20

Report

This manuscript presents a significant theoretical discovery regarding nonequilibrium systems modeled by Markov jump processes. I recommend this manuscript for publication in SciPost Physics, with optional revision. Please find the attached file for details.

Attachment

Recommendation

Publish (meets expectations and criteria for this Journal)

  • validity: high
  • significance: high
  • originality: top
  • clarity: good
  • formatting: good
  • grammar: good

Author:  Sara Dal Cengio  on 2025-08-15  [id 5732]

(in reply to Report 1 on 2025-05-20)
Category:
remark
question
answer to question

‘‘For nonequilibrium systems modeled by continuous-time Markov jump processes, graph-theoretic methods have proven to be useful for establishing thermodynamic structures and response theories. This manuscript presents a generalization of response relations in Markov jump processes, extending the authors’ earlier results for a single input current to the case of multiple input currents. The main conribution is a proof that the current along any edge of the network is given by a linear-affine function of the stationary currents along a selected set of edges (referred to as admissible edges). This linear-affine relation is further extended to non-stationary currents and holds in the Laplace domain, where a frequency variable conjugated to time is introduced. As consequences of the main result, the authors also show how Kirchhoff’s current law–a fundamental relation of network theory–is recovered, and how the linear-affine relation connects to the linear response relation at equilibrium. The mutual linearity of stationary currents in nonequilibrium networks was a fascinating discovery that, surprisingly, had not been observed prior to the authors’ earlier work (Ref. [33]). This manuscript presents a delicate and elegant graph-theoretic and algebraic treatment that identifies the general conditions under which the mutual linearity is preserved as a linear-affine relation, even when multiple input currents are involved. All the proofs and derivations are provided in sufficient detail, and the arguments are coherent and consistent with known results. Overall, the manuscript is clearly written and well organized. Given the generality of main result, which applies to arbitrary network topologies without assuming near-equilibrium conditions, I believe this manuscript will be of significant interest to researchers in nonequilibrium statistical physics and related fields. In conclusion, I recommend the manuscript for publication in SciPost Physics, with optional revision. I provide a few minor suggestions below that may help improve the clarity and accessibility of the manuscript. ’’

We agree with the referee’s summary of our work, and we thank the referee for the revision and the very positive evaluation.

‘‘The mathematical techniques and concepts used in the manuscript are not yet widely adopted among researchers in the field of nonequilibrium statistical physics. An illustration using a concrete example, such as the simple molecular motor considered in Ref. [33], would make the manuscript more accessible.’’

To help the reader grasp our results, we have added a new section 2.1 titled “Illustration” where we consider a simple graph, with a fixed set of two input currents and one output current. For several randomized sets of transition rates, we compute the output current and show that it always lies on a (hyper)plane in the space of input currents, identified by the mutual multilinearity relation (MML) [Eq. (7)]. We find this to be possibly the clearest illustration of MML and its geometrical implications. Other concepts, such as spanning trees, KCL and our notion of admissibility, are illustrated in the Introduction using the same graph, aiding the comprehension of the reader less familiar with graph-theoretic techniques.

‘‘The meaning of the final sentence in Sec. 5 is not sufficiently clear. Could the authors elaborate
on what is meant by the response relations in Eqs. (52) and (56) being “phenomenological”?
Providing a hypothetical experimental setup or measurement scheme where the relation could be
applied would enhance the clarity.’’

We agree that the choice of the term phenomenological is unfortunate. We changed it and now the sentence reads: “The physical relevance of our result is that it is operationally accessible: the currents’ susceptibilities do not have to be obtained analytically, but can be sampled from a realization of the process’’. What we mean is that the susceptibilities can be directly computed from equilibrium measurements of current covariances. This makes the relations practically applicable, for example, in experimental or numerical setups where fluctuations of the relevant currents can be recorded in equilibrium conditions. Since this is already improved, we opted not to include a hypothetical experimental setup to avoid digressing much, although we agree this is an interesting possibility.

‘‘As noted in the concluding remarks, the mutual multilinearity does not apply to arbitrary macroscopic input currents. It is restricted to cases where a set of admissible edges are perturbed. However, controlling such admissible edges may not always be feasible in practical applications. Could the authors comment on the practical relevance or possible advantages of the mutual multilinearity beyond its fundamental significance?’’

It is often unfeasible to control the rate of a single transition in isolation. In this context, the mutual multilinearity is practical because it extends the linearity result to arbitrarily sized input sets (provided that the set is admissible). We can identify several physical scenarios where this feature becomes relevant. To highlight this point, we have added the following sentence at the beginning of the Conclusion section: “Control over multiple transition rates is a scenario that emerges across disciplines. For example, tuning the concentration of chemicals in biochemical networks often affects several transition rates simultaneously. The same occurs when adjusting the temperature of a reservoir in a heat engine, modifying energy barriers in activated processes such as protein folding or ion transport, or altering transcription factors in gene expression dynamics. Understanding how stationary currents reorganize in response to changes in transition rates is therefore central”. As explained in the conclusion, if only a linear combination of currents can be measured—a macroscopic current—the relation is not that useful anymore. Nevertheless, it still carries the notion that the output current will respond linearly to input currents, and not exponentially for instance, which can be unexpected and a valuable insight in many cases.

---

## Round 1 · Referee Report · Anonymous (Referee 3) · 2025-5-27

Report

The authors prove a quite involved mutual multilinearity between edge currents of a Markov process on a finite graph. The output currents, i.e., those chosen to be observed, are probability currents flowing into some arbitrary edges of the graph. The input currents have transition rates (forward and backward) that can be controlled. They must also belong to a possible set of chords of the graph, although this is stated implicitly (see my first remark below).

The main result is that the output currents are linear-affine functions of the input currents, with coefficients that are only functions of the transition rates that are not controlled. This is crucial for the practical application of the main result since this means that those coefficients can be measured once and for all and then used to predict output currents for other input currents. Being not very readable, the main result is more important from its fundamental meaning and its practical applications than from its analytical form.

The graph-theoretic proof of the main result is easier to follow than the proof in Laplace space (I could not follow all the proof in Section 4). Moreover, the second proof is based on the existence of the linear coefficient between the derivative (with respect to transition rates) of the currents in Laplace space. Therefore, it does not provide the affine coefficient. Interestingly, I understand that the susceptibilities (linear coefficient) display a kind of independence: input edges can be studied individually while keeping their prediction ability for the output currents in the case of several input currents controlled simultaneously. This point is not clear to me from the perspective of section 3, when going from $n_c - 1$ to $n_c-2$ and by inspection of Eq. (23). Is this because removing edges from the graph (to obtain a reduced graph) is different from ensuring null currents on corresponding edges?

In conclusion, the manuscript presents rather complex results with potential practical applications in the long term. It fulfils all the acceptance criteria and should be published in Science Post after minor revision.

Requested changes

Remarks : - It seems that the input currents must always be chord currents, at least in the initialisation step of the first proof (this is what suggests the notation with $\alpha$ index). This is more restrictive than the admissible currents condition introduced by the authors. There is considerable freedom in choosing spanning trees, right. Edges not belonging to cycles are forbidden by the admissibility condition because they are on tree sub-parts of the graph and upon removal disconnect the graph (including the last edge of a branch since it lets a vertex alone, is'nt it?). What happens if one chooses for input currents two consecutive edges inside the very same cycle? The two currents cannot be controlled independently, so this situation does not make sense. However, this is not forbidden by the admissibility condition. Is admissibility a sufficient condition? In any case a precise definition of input and output currents would be welcome. Studies of often-occurring situations could help as well. - first paragraph of p6, the stationary probability is unique only for finite graphs (this is not stated in the above assumptions) - ''stoichiometric matrix'' appears several times instead of ''incidence matrix'' - ''Markov chain'' is sometimes used, but the authors have in mind ''continuous time Markov chain'' which should be named Markov jump process (no discrete time) - the justification of Eq. (18) is not evident to me. Could the author explain it in words or provide a simple example in an appendix? - page 9, last paragraph before the Remark; I am not yet convinced by the conclusion after Eq. (19): How can the linear and affine coefficients be independent of the rates for input edges? I understand that the potential $u$ depends only on the rates in the spanning tree by construction. However, the coefficients also depend on affinity $a_\alpha$ which depends on the rate in the input current. This is probably explained in Ref. [33], but I had not time to read this reference. - the conclusion mentions ''arbitrary network geometries''. The finite number of vertices should be mentioned here as well. - the last sentence of the conclusion is almost a repetition of a sentence five lines above. Making an exact repetition with just the word ''input'' replaced by ''output'' would be clearer to emphasise only this word. Otherwise, the reader must determine if the change at the end of the sentence that matters (which is not the case). (e.g.; remove ''multiple'' in the first sentence and change the last sentence to ''[...] output currents are a linear combination of edge currents.'')

Recommendation

Publish (easily meets expectations and criteria for this Journal; among top 50%)

  • validity: high
  • significance: high
  • originality: high
  • clarity: good
  • formatting: good
  • grammar: excellent

Author:  Sara Dal Cengio  on 2025-08-15  [id 5736]

(in reply to Report 3 on 2025-05-27)

“The authors prove a quite involved mutual multilinearity between edge currents of a Markov process on a finite graph. The output currents, i.e., those chosen to be observed, are probability currents flowing into some arbitrary edges of the graph. The input currents have transition rates (forward and backward) that can be controlled. They must also belong to a possible set of chords of the graph, although this is stated implicitly (see my first remark below).

The main result is that the output currents are linear-affine functions of the input currents, with coefficients that are only functions of the transition rates that are not controlled. This is crucial for the practical application of the main result since this means that those coefficients can be measured once and for all and then used to predict output currents for other input currents. Being not very readable, the main result is more important from its fundamental meaning and its practical applications than from its analytical form.

The graph-theoretic proof of the main result is easier to follow than the proof in Laplace space (I could not follow all the proof in Section 4). Moreover, the second proof is based on the existence of the linear coefficient between the derivative (with respect to transition rates) of the currents in Laplace space. Therefore, it does not provide the affine coefficient.”

We agree with the summary of our work made by the referee and we are grateful for the careful read. We respectfully disagree with the sentence “[the second proof] does not provide the affine coefficient”. Proving Eq. (27) is fully equivalent to proving Eq. (28), as we stated at the beginning of Section 4.2. Using the resolving method, we prove Eq. (28), from which the linear-affine formulation of Eq. (27) follows directly. Then, the affine coefficient is simply given by the (Laplace) current for a reduced graph where the input edges are removed. We have added a sentence about the affine coefficient at the end of Section 4.2.

“Interestingly, I understand that the susceptibilities (linear coefficient) display a kind of independence: input edges can be studied individually while keeping their prediction ability for the output currents in the case of several input currents controlled simultaneously. This point is not clear to me from the perspective of section 3, when going from nc- 1 to nc- 2 and by inspection of Eq. (23). Is this because removing edges from the graph (to obtain a reduced graph) is different from ensuring null currents on corresponding edges?”

We thank the referee for raising this point. Removing edges in the reduced graph is indeed fully equivalent to ensuring null currents on the corresponding edges (or, in other words, send both rates to zero). We agree that the connection between the inductive step of Section 3 and the “independence” among susceptibilities discussed at the end of Section 4 was not made explicit. We have now added a remark at the end of Section 3.2 clarifying this link: the independence among susceptibilities is already contained in Eq. (24) when the appropriate rates are sent to zero. We have added a remark about this in the third paragraph of the conclusion and perspectives.

“In conclusion, the manuscript presents rather complex results with potential practical applications in the long term. It fulfils all the acceptance criteria and should be published in Science Post after minor revision.”

We thank the referee for such a positive evaluation of our work.

“Remarks : - It seems that the input currents must always be chord currents, at least in the initialisation step of the first proof (this is what suggests the notation with α index). This is more restrictive than the admissible currents condition introduced by the authors. There is considerable freedom in choosing spanning trees, right. Edges not belonging to cycles are forbidden by the admissibility condition because they are on tree sub-parts of the graph and upon removal disconnect the graph (including the last edge of a branch since it lets a vertex alone, is'nt it?). What happens if one chooses for input currents two consecutive edges inside the very same cycle? The two currents cannot be controlled independently, so this situation does not make sense. However, this is not forbidden by the admissibility condition. Is admissibility a sufficient condition? In any case a precise definition of input and output currents would be welcome.”

The admissibility condition is both necessary and sufficient for the result to apply to a given set of input currents. It is fully equivalent to stating that the corresponding input edges can be chosen as chords within a (non-unique) fundamental set. In other words, there exists (at least) one spanning tree not containing any of the input edges. A fundamental set of chords is a maximal set of n_c edges whose removal does not disconnect the graph—that is, whose removal forms a spanning tree. In the manuscript, after Eq. (6), we have added a sentence: “These edges are known as chords and they form what is known as the fundamental set”. Saying that a set of input currents is admissible is saying that their removal does not disconnect the graph (see Fig. 3). The cardinality of the input set is free to vary from 1 to n_c. If it equals n_c, the set of input edges forms a fundamental set. We hope this clarifies the point to the reader.
Regarding the specific case raised by the referee—choosing two consecutive edges in the same cycle—we note that if the removal of these two edges does not disconnect the graph (i.e., does not isolate any vertex), then this is a valid choice of input edges. In that case, they can also be chosen as chords, with two different cycles associated with them. We emphasize that the choice of cycles is not independent of the choice of chords. Specifically, the definition of cycles depend on the non-unique choice of spanning tree; thus, two input edges belonging to the same cycle for a given choice of spanning tree might belong to different cycles for a different spanning tree.

“In any case a precise definition of input and output currents would be welcome. Studies of often-occurring situations could help as well.”

We have added a sentence after Eq. (6): “In this work, we generalize Eq. (6) to the case where the n < nc currents are a subset of some fundamental set, a condition that we term admissibility (see Fig. 3). These n currents ȷ_i are termed input currents, while any other current ȷ_e in the graph can be regarded as output.”. Concerning often-occurring situations we have added a sentence referring to real-system scenarios at the beginning of the Conclusion: “Control over multiple transition rates is a scenario that emerges across disciplines. For example, tuning the concentration of chemicals in biochemical networks often affects several transition rates simultaneously. The same occurs when adjusting the temperature of a reservoir in a heat engine, modifying energy barriers in activated processes such as protein folding or ion transport, or altering transcription factors in gene expression dynamics. Understanding how stationary currents reorganize in response to changes in several transition rates is therefore central.”

“First paragraph of p6, the stationary probability is unique only for finite graphs (this is not stated in the above assumptions) - ''stoichiometric matrix'' appears several times instead of ''incidence matrix'' - ''Markov chain'' is sometimes used, but the authors have in mind ''continuous time Markov chain'' which should be named Markov jump process (no discrete time)”

We agree with the referee’s comments and we have unified the terminology of the manuscript using “incidence matrix”, “continuous-time Markov chains” and “ finite network geometries ”.

“The justification of Eq. (18) is not evident to me. Could the author explain it in words or provide a simple example in an appendix?”

Eq. (19) [previously (18)] is obtained simply by replacing Eq. (18) [previously (17)] into the condition of normalization, which appears on the first line of page 10. Eq. (18) in turn comes from Eq. (13) and we refer to Appendix A for the connection between affinities integrated along unique paths and potentials.

“page 9, last paragraph before the Remark; I am not yet convinced by the conclusion after Eq. (19): How can the linear and affine coefficients be independent of the rates for input edges? I understand that the potential u depends only on the rates in the spanning tree by construction. However, the coefficients also depend on affinity a_α which depends on the rate in the input current. This is probably explained in Ref. [33], but I had not time to read this reference.” 

The logic of the inductive seed is the following: We begin with a fundamental set of n_c input edges for which KCL ensures that any output current on the spanning trees is a linear combination of chord currents with algebraic coefficients. We then pick a chord alpha from this fundamental set and remove it from the input set, effectively reclassifying it as an output edge. The goal is to show that the current on edge α does not depend on the rates of the remaining n_c-1 input currents. The referee is correct in saying that the affinity a_α depends on the rate of the edge α. However, since α has been removed from the input set, its current is no longer treated as an input—it is now part of the output. The reduction in the number of input currents is precisely the point of the inductive step. We also recall that, in Eq. (20), c and P are topological u, K and k^α only depend on the rates of the spanning tree, a_α depends on the rates of the tree and of the (now output) edge α.

“the conclusion mentions ''arbitrary network geometries''. The finite number of vertices should be mentioned here as well.” 

We agree with the referee and have performed the change.

“the last sentence of the conclusion is almost a repetition of a sentence five lines above. Making an exact repetition with just the word ''input'' replaced by ''output'' would be clearer to emphasise only this word. Otherwise, the reader must determine if the change at the end of the sentence that matters (which is not the case). (e.g.; remove ''multiple'' in the first sentence and change the last sentence to ''[...] output currents are a linear combination of edge currents.’')” 

We agree with the referee. The last sentence of the manuscript now reads: “As pointed out in Ref. [33], however, the result easily extends to the case of macroscopic output currents that are a linear combination of multiple edge currents.”

---

## Round 1 · Referee Report · Anonymous (Referee 2) · 2025-5-27

Report

While we do no question the validity of the results presented by the authors, we also note that several aspects in the introductory text as well as in the motivation are vague, imprecise or unclear. We believe that the authors should address these issues before the manuscript can be recommended for publication.

  • Equations 1 and 6 are the same equation, with slightly different notation and with different amounts of detail in the accompanying text. This is rather confusing and can be easily remedied by removing one of the two copies and concentrating all the information in one place.

  • Page 3: "According to Kirchhoff current law, any stationary edge current ȷ_e can be expressed as a superposition of the stationary currents ȷi flowing along the chords of a fundamental set". This is not what Kirchhoff's current law states. In fact, Kirchhoff's current law states that the algebraic sum of the currents at a node is zero. Here and elsewhere in the manuscript, the authors explain specific results invoking Kirchhoff's current law (KCL in the text). This is very confusing, it makes it very difficult to assess the validity of the authors' statements, in particular the assumptions that they work under. The authors should be more precise about what is a consequence of KCL, the Gutman Mallion Essam theorem, and the Cycle Theorem (as in ref 37)

  • Still about Eq 1: "According to Kirchhoff current law, any stationary edge current ȷ_e can be expressed as a superposition of the stationary currents ȷ_i flowing along the chords of a fundamental set, as [Eq. 1]" This statement is in general wrong. Saying "any stationary edge current ȷ_e" also means edges outside of cycles. According to the statements above, stationary currents on edges outside of cycles are zero, and this is only true for very specific choices of boundary conditions.

  • Related to the above, the authors never mention boundary conditions, at least not explicitly. Note however that the uniqueness of stationary states is guaranteed only after boundary values are chosen. Our understanding is that the authors follow the formalism of Schnakenberg, where affinities generate currents in cycles. Notice that when a current is externally induced in a cycle edge, that edge is in the boundary of the system. Under these assumptions, we would certainly agree with the above statement about all the j_e, but the authors should be clearer and more rigorous about it.

  • After Eq 6, the incidence matrix is called stoichiometric matrix. We recommend that the authors stick to the standard "incidence matrix".

  • In regard to the general premise of the work, and in particular the need to prove linearity in response to "input currents" (as they are called in the manuscript). The equations governing stationary states and time evolution are linear in the choice of boundary values and a reader might hypothesize that linear response to external/boundary forces is a trivial consequence. We invite the authors to clearly state and exemplify why this result is not trivial and why it indeed requires a proof. While we do no question the validity of the results presented by the authors, we also note that several aspects in the introductory text as well as in the motivation are vague, imprecise or unclear. We believe that the authors should address these issues before the manuscript can be recommended for publication.

  • Equations 1 and 6 are the same equation, with slightly different notation and with different amounts of detail in the accompanying text. This is rather confusing and can be easily remedied by removing one of the two copies and concentrating all the information in one place.

  • Page 3: "According to Kirchhoff current law, any stationary edge current ȷ_e can be expressed as a superposition of the stationary currents ȷi flowing along the chords of a fundamental set". This is not what Kirchhoff's current law states. In fact, Kirchhoff's current law states that the algebraic sum of the currents at a node is zero. Here and elsewhere in the manuscript, the authors explain specific results invoking Kirchhoff's current law (KCL in the text). This is very confusing, it makes it very difficult to assess the validity of the authors' statements, in particular the assumptions that they work under. The authors should be more precise about what is a consequence of KCL, the Gutman Mallion Essam theorem, and the Cycle Theorem (as in ref 37)

  • Still about Eq 1: "According to Kirchhoff current law, any stationary edge current ȷ_e can be expressed as a superposition of the stationary currents ȷ_i flowing along the chords of a fundamental set, as [Eq. 1]" This statement is in general wrong. Saying "any stationary edge current ȷ_e" also means edges outside of cycles. According to the statements above, stationary currents on edges outside of cycles are zero, and this is only true for very specific choices of boundary conditions.

  • Related to the above, the authors never mention boundary conditions, at least not explicitly. Note however that the uniqueness of stationary states is guaranteed only after boundary values are chosen. Our understanding is that the authors follow the formalism of Schnakenberg, where affinities generate currents in cycles. Notice that when a current is externally induced in a cycle edge, that edge is in the boundary of the system. Under these assumptions, we would certainly agree with the above statement about all the j_e, but the authors should be clearer and more rigorous about it.

  • After Eq 6, the incidence matrix is called stoichiometric matrix. We recommend that the authors stick to the standard "incidence matrix".

  • In regard to the general premise of the work, and in particular the need to prove linearity in response to "input currents" (as they are called in the manuscript). The equations governing stationary states and time evolution are linear in the choice of boundary values and a reader might hypothesize that linear response to external/boundary forces is a trivial consequence. We invite the authors to clearly state and exemplify why this result is not trivial and why it indeed requires a proof.

Recommendation

Ask for minor revision

  • validity: good
  • significance: high
  • originality: good
  • clarity: low
  • formatting: good
  • grammar: excellent

Author:  Sara Dal Cengio  on 2025-08-15  [id 5733]

(in reply to Report 2 on 2025-05-27)
Category:
remark
question
answer to question
objection
reply to objection

“While we do no question the validity of the results presented by the authors, we also note that several aspects in the introductory text as well as in the motivation are vague, imprecise or unclear. We believe that the authors should address these issues before the manuscript can be recommended for publication.”

We thank the referee for the review and constructive comments, which we believe were satisfactorily addressed in the resubmitted version of the manuscript, improving its clarity.

“Equations 1 and 6 are the same equation, with slightly different notation and with different amounts of detail in the accompanying text. This is rather confusing and can be easily remedied by removing one of the two copies and concentrating all the information in one place.”

Indeed the equations are the same, and simultaneously looking at them can be confusing; however, we believe the chosen notation enhances the readability and logical flow of the paper, thus we have deliberately kept them both because they serve different purposes. Equation (1) in the introduction omits the set of edges \mathcal{E} and the symbol for a spanning tree \mathcal{T}, since we do not want to overload the introduction with definitions. After Eq. (6), we have added “Notice this is equivalent to Eq. (1)”, and in fact there is some value in comparing both equations to highlight that the set \mathcal{E} minus \mathcal{T} is equivalent to the set of chords from 1 to n_c.

“Page 3: "According to Kirchhoff current law, any stationary edge current ȷ_e can be expressed as a superposition of the stationary currents ȷi flowing along the chords of a fundamental set". This is not what Kirchhoff's current law states. In fact, Kirchhoff's current law states that the algebraic sum of the currents at a node is zero. Here and elsewhere in the manuscript, the authors explain specific results invoking Kirchhoff's current law (KCL in the text). This is very confusing, it makes it very difficult to assess the validity of the authors' statements, in particular the assumptions that they work under. The authors should be more precise about what is a consequence of KCL, the Gutman Mallion Essam theorem, and the Cycle Theorem (as in ref 37).”

For clarity, we have changed the formulation in Page 3 to which the referee refers to: “As a direct consequence of Kirchhoff current law […]”. Indeed, we agree with the referee that KCL states that the algebraic sum of the currents at every node is zero at stationarity. This is what Eq. (5) in the manuscript expresses. In the new version of the manuscript, we wrote [just before Eq. (5)]: “A straightforward consequence of the master equation is Kirchhoff’s Current Law (KCL)[..] stating that the in-and-out stationary currents balance at any vertex x, locally. As we now detail, there also exists a global version of KCL, equivalent to Eq. (5), expressed by means of spanning trees, a central notion of graph theory.” To avoid confusion, we also added: “In the following, we refer to Eq. (6) as KCL, as it directly follows from Kirchhoff’s Current Law and is sometimes used as an alternative representation of it in the literature”. Finally, we added a footnote just before Eq. (5) commenting on the equivalence between the two formulations of KCL. We thank the referee for spotting this imprecision.

“Still about Eq 1: "According to Kirchhoff current law, any stationary edge current ȷ_e can be expressed as a superposition of the stationary currents ȷ_i flowing along the chords of a fundamental set, as [Eq. 1]" This statement is in general wrong. Saying "any stationary edge current ȷ_e" also means edges outside of cycles. According to the statements above, stationary currents on edges outside of cycles are zero, and this is only true for very specific choices of boundary conditions.”

As we consider a continuous-time Markov chain (on a finite network) with conserved probability, the boundary conditions are the simplest one: there are no imposed boundary currents or probabilities. We stress that we only focus on closed systems, meaning that (in the language of chemical reaction networks) no chemostatting or external reservoirs are considered. In this context, non-zero stationary currents only arise due to the violation of Kolmogorov’s criterion along the cycles of the network. In the absence of “emergent cycles” induced by chemostatting (i.e., in open systems), edges that do not belong to any internal cycle indeed carry zero stationary current.

“Related to the above, the authors never mention boundary conditions, at least not explicitly. Note however that the uniqueness of stationary states is guaranteed only after boundary values are chosen. Our understanding is that the authors follow the formalism of Schnakenberg, where affinities generate currents in cycles. Notice that when a current is externally induced in a cycle edge, that edge is in the boundary of the system. Under these assumptions, we would certainly agree with the above statement about all the j_e, but the authors should be clearer and more rigorous about it.”

Since we only consider closed systems, meaning that no currents are externally induced in our framework, the boundary conditions are very simple: there are no imposed currents or probabilities. To highlight this point, we added a sentence after Eq. (3): "The absence of source or sink terms in Eq. (3) ensures probability conservation.” This is why in the perspectives we mention: “Extending our results to the framework of open (driven) systems is an interesting direction. In the case of a single input current we have proved that the linearity extends to the case of open systems despite the fact that global conservation of probability breaks down [33]”.

“After Eq 6, the incidence matrix is called stoichiometric matrix. We recommend that the authors stick to the standard "incidence matrix”.”

We thank the referee for spotting this inconsistency. In the new version of the manuscript, we use only the term “incidence matrix”.

“In regard to the general premise of the work, and in particular the need to prove linearity in response to "input currents" (as they are called in the manuscript). The equations governing stationary states and time evolution are linear in the choice of boundary values and a reader might hypothesize that linear response to external/boundary forces is a trivial consequence. We invite the authors to clearly state and exemplify why this result is not trivial and why it indeed requires a proof.”

We would like to stress that here we do not discuss linearity with respect to external forces and/or boundary currents, but linearity with respect to the modification of several transition rates of the Markov chain. It is true that the Markovian time evolution in Eq. (3) is linear in the probabilities, as much as Eq. (5) is linear in the stationary currents. Yet, the dependency of the stationary probability [and consequently of the stationary currents, see Eq. (4)] on the transition rates is non-linear and intricate, as can be seen directly from the Markov chain tree theorem. This is why the mutual multilinearity is a nontrivial result and requires a convoluted proof.

---

## Editorial Decision

resubmitted